# Citizen science provides a reliable and scalable tool to track disease-carrying mosquitoes

John R.B. Palmer [1,2,3], Aitana Oltra [1,3], Francisco Collantes [4], Juan Antonio Delgado[4], Javier Lucientes[5], Sarah Delacour[5], Mikel Bengoa[5], Roger Eritja[3] & Frederic Bartumeus [1,3,6]

Recent outbreaks of Zika, chikungunya and dengue highlight the importance of better understanding the spread of disease-carrying mosquitoes across multiple spatio-temporal scales. Traditional surveillance tools are limited by jurisdictional boundaries and cost constraints. Here we show how a scalable citizen science system can solve this problem by combining citizen scientists' observations with expert validation and correcting for sampling effort. Our system provides accurate early warning information about the Asian tiger mosquito (*Aedes albopictus*) invasion in Spain, well beyond that available from traditional methods, and vital for public health services. It also provides estimates of tiger mosquito risk comparable to those from traditional methods but more directly related to the human–mosquito encounters that are relevant for epidemiological modelling and scalable enough to cover the entire country. These results illustrate how powerful public participation in science can be and suggest citizen science is positioned to revolutionize mosquito-borne disease surveillance worldwide.

[1] Centre d'Estudis Avançats de Blanes (CEAB-CSIC), Blanes 17300, Spain. [2] Universitat Pompeu Fabra, Barcelona 08005, Spain. [3] CREAF, Cerdanyola del Vallès 08193, Spain. [4] Universidad de Murcia, Murcia 30100, Spain. [5] Universidad de Zaragoza, Zaragoza 50013, Spain. [6] ICREA, Institut Catala de Recerca i Estudis Avançats, Barcelona 08010, Spain. Correspondence and requests for materials should be addressed to J.R.B.P. (email: john.palmer@upf.edu) or to F.B. (email: fbartu@ceab.csic.es)

Invasive species that are also disease vectors play large and growing roles in driving environmental change and threatening public health[1, 2]. Climate change, the proliferation of global trade routes and transportation technologies, and the growth in human mobility have meant that non-native species are increasingly introduced into new areas in which they are able to thrive, while native species increasingly push the boundaries of their ranges, all with implications for biodiversity, extinction, ecosystem disruption and economic loss[3–5]. When these new invaders also carry human diseases, their impacts balloon to include altering pathogen transmission cycles and directly threatening human health[1, 2, 6], as recent and ongoing outbreaks of Zika, chikungunya and dengue demonstrate[7, 8].

The study of invasive disease vectors is of critical importance for these reasons. It is often constrained, however, by questions of scale. The same factors that make species invasions increasingly prevalent also ensure that their analysis requires observation across a large range of scales, from global to local in terms of space; from centuries to minutes in terms of time[1, 3, 9]. Traditional tools for monitoring and control are often impossible to implement at the larger and smaller ends of this range of scales, as dispersal through commercial shipping, airline traffic, motor vehicles flows and other human activities allow invasive species to quickly jump past jurisdictional boundaries and financially constrained study sites, and as budgets for long-term and fine-grained surveillance are cut to meet immediate needs[10, 11]. What is needed is some mechanism providing the potential for massive and sustained collaboration in data collection and analysis—a mechanism that scales with the same flexibility as the invasive species in question.

In fact, such a mechanism is among the core benefits promised by the emerging paradigm of networked citizen science. Public participation in science, which dates back thousands of years[12–14], has entered a new phase as citizen scientists are now being enlisted and linked on previously unimaginable scales by the Internet, inexpensive mobile phones with powerful sensor arrays, and other modern information and communications technologies[15, 16]. These new networks of citizen scientists are well positioned to study species invasions around the world without hitting the geographic barriers and economic constraints of traditional methods. We demonstrate this with Mosquito Alert, a scalable citizen science system that we developed to tackle the problem of Asian tiger mosquitoes (*Aedes albopictus*) in Spain[17, 18].

The tiger mosquito is a potential vector of at least 22 arboviruses, including Zika, dengue and chikungunya. It breeds in man-made containers, thrives in urban areas, and has spread from the western Pacific and Southeast Asia to Europe, Africa, the Middle East and the Americas over the past three decades[19]. It was first detected and reported in Spain in 2004[20]. It has since become established along Spain's eastern coast[11, 17, 18], where it is well known for its aggressive daytime biting, which degrades the quality of life, harms summer tourism and presents a serious epidemiological risk to which public health authorities at local, regional, national and European levels have been attempting to respond[1, 8, 21].

As with other invasive species, the tiger mosquito invasion shows a stratified dispersal strategy, combining small-scale diffusion with long-distance dispersal, resulting in the formation of isolated colonies that greatly increase the spread[22]. With the tiger mosquito, these broader dispersal scales are apparently facilitated by commercial shipping, airline traffic and other human transportation[11, 20–23]. At small scales, tiger mosquito prevalence and disease transmission risks vary considerably based on complex interactions and feedbacks related to socio-ecological factors[24–27]. Traditional tiger mosquito surveillance generally involves trapping the mosquito's eggs using ovitraps, small containers of water that must be set and recursively checked by experts in the field. The tiger mosquito's fast-moving and jumpy invasion, combined with tight budgets, however, has allowed it to frequently outpace ovitrap surveillance, with initial detections in new areas coming instead from the general public[11, 20, 28, 29]. That is consistent with the literature suggesting that early warning and invasion-front mapping are tasks for which citizen science is especially well suited[30]. Moreover, the process of public detection is facilitated by the tiger mosquito's distinctive appearance and aggressive biting[11, 20, 21], making it simultaneously easy to recognize and difficult to ignore.

Mosquito Alert capitalizes on these characteristics, encouraging the public to systematically report tiger mosquito sightings, facilitating collaboration between citizen scientists, researchers, and public health administrations, and raising awareness about steps everyone can take to reduce the risk of mosquito-borne diseases. The goal is to generate an efficient system for research, surveillance and control. This has involved outreach and training to help citizen scientists (more than 38,400 registered to date, with thousands of active participants at any given time during the mosquito season) accurately identify and report tiger mosquitoes, community engagement strategies to encourage reporting, and expert–citizen collaboration, whereby a team of entomologists reviews and validates all reports that include photographs[31].

The important questions, of course, are how citizen science differs from traditional methods and whether it can provide information of comparable quality. To answer these questions, we have collected data from two radically different surveillance methods operating simultaneously across a large territory. We have combined nearly 5000 Mosquito Alert reports from citizen scientists with data on nearly every ovitrap monitored in Spain during 2014–2015 (over 1500 traps, checked every ~2 weeks). We compare the citizen science data with the

**Table 1 Comparison of knowledge added during 2014–2015 about tiger mosquito range in Spain based on the two main surveillance methodologies deployed in the country: ovitraps (which capture mosquito eggs) and citizen science (Mosquito Alert)**

| Surveillance method 2014–2015 | New municipalities detected | | Area of newly detected municipalities | | Near distance to invasion front | |
|---|---|---|---|---|---|---|
| | No. | % | km$^2$ | % | Mean km | Median km |
| Citizen science alone | 108 | 39% | 5761 | 34% | 37 | 17 |
| Citizen science inclusive | 175 | 64% | 10,948 | 65% | 43 | 16 |
| Ovitraps alone | 99 | 36% | 5787 | 35% | 20 | 0 |
| Ovitraps incl. inclusive | 166 | 61% | 10,975 | 665 | 34 | 0.3 |
| Total | 274 | 100% | 16,736 | 100% | 35 | 6 |

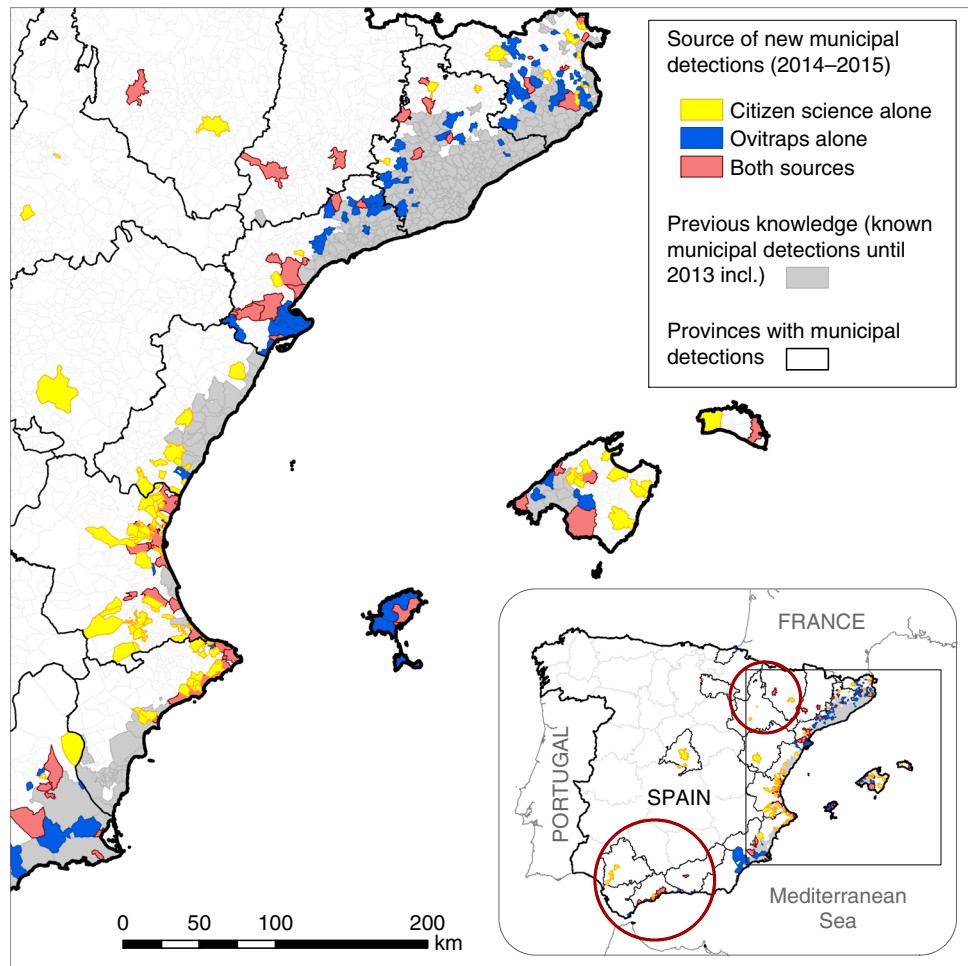

**Fig. 1** Range of expansion of the tiger mosquito in Spain (Canary Islands, Ceuta and Melilla excluded) in 2014–2015. Provinces with municipal-level detections are highlighted. Grey areas denote state of knowledge at the municipal level based on ovitrap surveillance through 2013 (source data from ref. 17). Other colours indicate municipal detections during 2014–2015 by expert validated reports from citizen science (yellow), ovitrap surveillance (dark blue) and both methods (light red) (source data from refs. 17, 18). Red circles indicate areas far from the main invasion front, from which the discoveries of the species were corroborated in the field using ovitraps, but triggered by citizen science[32, 33]. Boundary data from Spanish National Geographic Institute[53]. © Instituto Geográfico Nacional

traditional ovitrap data in terms of (1) economic cost, (2) effectiveness as an early warning mechanism and (3) ability to capture spatio-temporal variation in human–mosquito encounter probabilities. We find that citizen science costs less than traditional methods and provides early warning information and human–mosquito encounter probabilities of comparable quality with larger geographical coverage.

## Results

**Economic cost**. Our first point of comparison is cost. Mosquito Alert operated during 2014–2015 with a total budget of 300,000 Euros, covering 487,775 km$^2$ in Spain for an average cost of about 1.23 Euros per km$^2$ per month. In contrast, we estimate that the much more labour-intensive ovitraps, which must be set and checked by experts in the field and lab, cost about 9.36 Euros per km$^2$ per month (see Methods for detailed calculation)—nearly eight times the cost of Mosquito Alert. Moreover, these ovitrap costs are mostly recurring labour expenses, which should scale linearly with area and time. In contrast, the Mosquito Alert costs are mostly associated with community building and outreach and non-recurring investments in technology, both aimed at maintaining and attracting increasing numbers of participants year after year.

**Early warning**. Table 1 and Figs. 1–3 compare the methods credited with the first detections of tiger mosquitoes in Spanish municipalities during 2014–2015. We observe that Mosquito Alert accounts for first detections far beyond the known invasion area, to which traditional surveillance methods are usually limited (Fig. 1). The known invasion area comprises the municipalities in which the mosquito has already been detected, and ovitraps are usually deployed inside this area or in municipalities contiguous with its edges (the invasion front). The Mosquito Alert detections of tiger mosquitoes far beyond this area demonstrate that the invasion process can occur in jumps.

The comparison between Mosquito Alert and ovitraps is complicated somewhat by the fact that Mosquito Alert detections in new municipalities (i.e., first detections) triggered ovitrap deployment in some of these municipalities[32, 33]. We know that this occurred in at least 11 cases[32, 33], covering the regions of Andalusia (5)[26], Murcia (4), Aragon (1)[27] and Catalonia (1), and we suspect others in Valencia and the Balearic Islands. Quantifying this is difficult because new Mosquito Alert detections are made public and also reported to the Centre for the Coordination of Health Warnings and Emergencies (CCAES), which in turn notifies the regional public health actors. The latter must investigate tiger mosquito reports in the field in order to

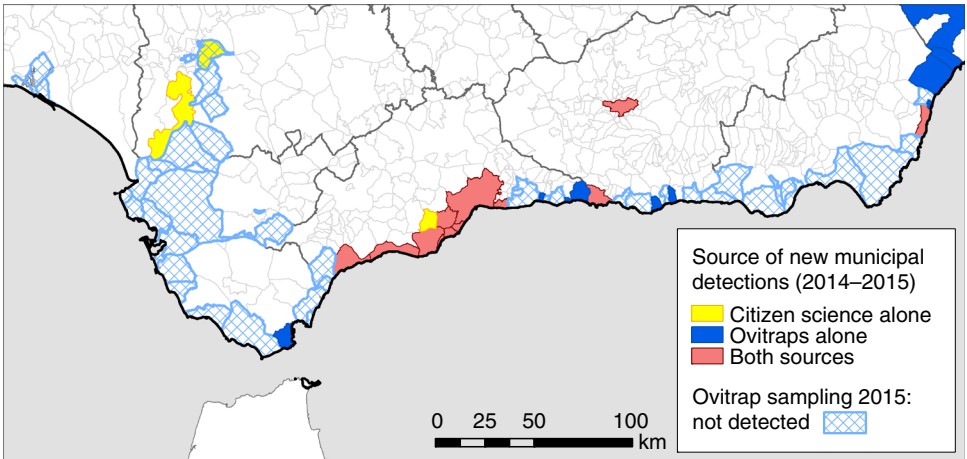

**Fig. 2** New detections based on the ovitrap and Mosquito Alert tiger mosquito surveillance performed in the south of Spain in 2014–2015 (as in Fig. 1). Municipalities surveilled with ovitraps during 2015 but ending in no-detections are shown in light dashed blue (source data from ref. 18). Note that a lot of sampling effort is unsuccessful (light dashed blue). Successful detections come from ovitraps alone (dark blue), citizen scientists alone (yellow) or both ovitraps and citizen scientists (light red). In 4 out of the 12 red municipalities, ovitrap surveillance was triggered by citizen science alerts through the Mosquito Alert platform but citizen science reports cannot always be confirmed in the field (e.g. ovitraps are not always placed in municipalities where citizen science reports suggest presence; even when they are placed in these municipalities low ovitrap or population densities may generate false negatives). Boundary data from Spanish National Geographic Institute[53]. © Instituto Geográfico Nacional

officially confirm the species' presence, and this is usually done by deploying ovitraps. If presence is confirmed, mosquito public health protocols and management actions are activated, and the new discovery is often reported in scientific publications[32, 33], but the source of the initial information that led the regional public health actors to deploy ovitraps is not always clear. Figure 2 illustrates how this process played out in 2015 in the south of Spain, after Mosquito Alert first detected tiger mosquitoes there the previous year[32]. In Andalusia, ovitraps were deployed in five municipalities from which reliable Mosquito Alert reports had originated, but also in many other municipalities along the coastline (Fig. 2). In four out of these five municipalities, tiger mosquito presence was confirmed in the field by ovitraps. Given this connection between Mosquito Alert reports and ovitrap deployment, we distinguish, in the following analysis, between the non-overlapping sets of municipalities in which ovitraps or Mosquito Alert observations are alone responsible for the detection and the overlapping sets of total municipalities for which each source is credited.

Of 274 Spanish municipalities in which tiger mosquitoes were detected for the first time in 2014–2015, Mosquito Alert is alone credited with detections in 108 (39%). In total (including the municipalities in which both sources are credited), Mosquito Alert is credited with detections in 175 municipalities (64%). In contrast, ovitraps are alone credited in 99 municipalities (36%), with a total (including municipalities in which both are credited) of 166 (61%). The municipalities for which Mosquito Alert is alone credited and those for which ovitraps are alone credited cover approximately the same area: 5761 and 5787 km², respectively, as do the totals for each source: 10,948 and 10,975 km² (Table 1).

That Mosquito Alert detections tend to be farther from the known invasion area than ovitrap detections (as noted above and visible in Fig. 1) is clear from an analysis of the newly detected municipalities. Of the municipalities for which Mosquito Alert is alone credited, only 19% lie along the border of the known invasion area. Of the total Mosquito Alert municipalities, only 25% lie along this border. The rest are separated from the known invasion area by at least one intervening municipality. In contrast, 60% of the municipalities for which ovitraps are alone credited

and 50% of the total ovitrap municipalities lie along the border of the known invasion area. A simple Chi-squared test of the equality of these proportions against the alternative that the ovitrap proportion is greater yields $p$-values well below 0.01, regardless of whether we use the non-overlapping or the overlapping sets of distances. Thus, the observed differences are very unlikely to be the result of random variation alone.

The greater distance of Mosquito Alert detections from the known invasion area can be further quantified if we compare median and mean distances from the borders of the detected municipalities to the nearest borders of the known invasion area. The median and mean distances of the municipalities for which Mosquito Alert is alone credited are 17 and 37 km, while those of the municipalities for which ovitraps are alone credited are 0 km (i.e., contiguous) and 20 km. If we include the overlapping municipalities in these calculations, the Mosquito Alert median drops only slightly, to 16 km, and its mean rises to 43 km, while the ovitrap median remains <0.5 km and its mean rises to 34 km (Table 1). The difference between these distributions is obvious in Fig. 3 and strongly supported by Mann–Whitney tests, which yield $p$-values well below 0.01 regardless of whether overlapping municipalities are included. The geographic scale of the Mosquito Alert detections is especially apparent in the new detections located far to the south and west of the known invasion front (Fig. 1)[11, 33] and the differences would be even more pronounced had Mosquito Alert detections not triggered overlapping ovitrap detections.

We are confident that nearly all of the Mosquito Alert detections are true positives based on a comparison with municipalities in which ovitraps were deployed contemporaneously: of 125 municipalities in which ovitraps were deployed and failed to detect tiger mosquitoes in 2015, only 4 were classified as positive by Mosquito Alert. This gives a specificity (true negative rate) of 97%, assuming that ovitraps have perfect sensitivity, and higher if the ovitraps failed to detect true positives in any of these municipalities. This high specificity makes sense, given that the Mosquito Alert early warning system relies only on citizen scientists' observations that include photographs that are subsequently validated by experts (see Methods).

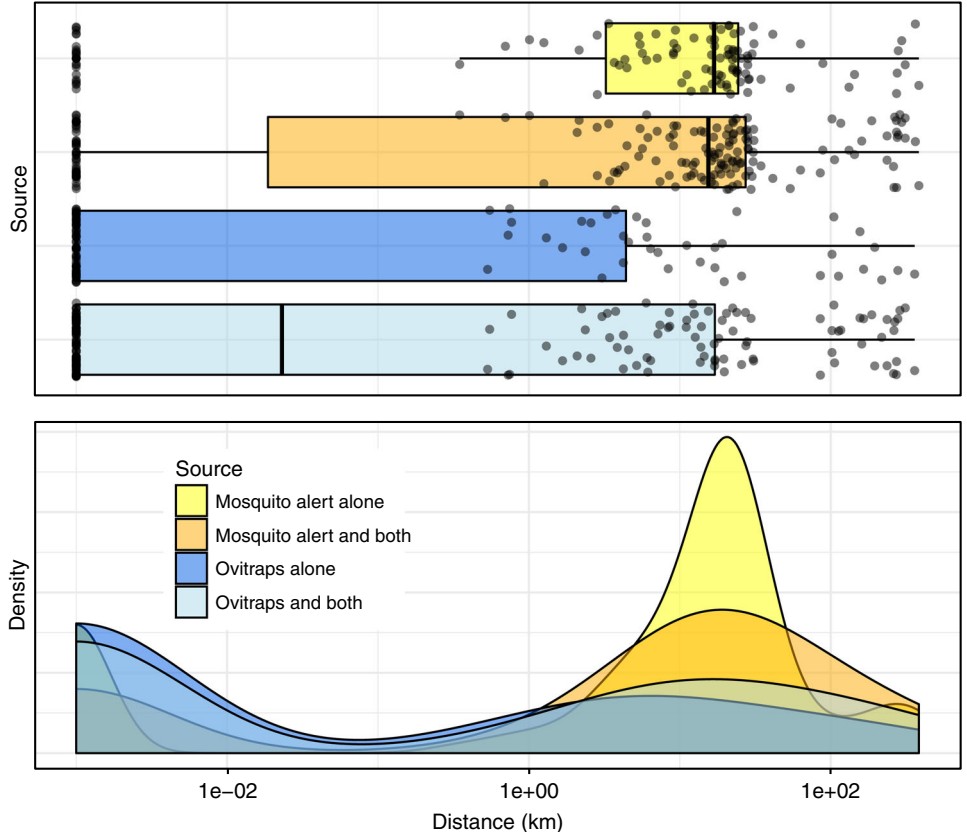

**Fig. 3** Distributions of distances between known tiger mosquito invasion front and new municipalities in which tiger mosquitoes were detected in 2014–2015. New municipalities are divided into those in which Mosquito Alert is alone credited for the detection (yellow), those in which Mosquito Alert is alone credited combined with those in which both Mosquito Alert and ovitraps are credited (in other words, all municipalities in which Mosquito Alert is credited; orange), those in which ovitraps are alone credited (dark blue), and those in which ovitraps are alone credited combined with those in which both are credited (in other words, all those in which ovitraps are credited; light blue). All distances calculated between municipality boundaries, using shortest distance to invasion front, and shown on x axis using log-scale to improve visualization of long tails (1 m added to all 0-km distances). Top: boxplots with boxes encompassing central 50% of the data and central bars indicating medians. Whiskers extend to 1.5 times the inter-quartile ranges. Raw data shown with jittered black points. Bottom: Gaussian kernel density estimates

We also use the comparison with ovitraps to estimate sensitivity. Of 112 municipalities in which ovitraps were deployed and detected tiger mosquitoes in 2015, 53 were classified as positive by Mosquito Alert, giving a sensitivity (true positive rate) of 47%. This figure must be interpreted in light of Mosquito Alert's large geographic coverage and the general problem of surveillance sensitivity at the edges of invasion fronts, where colonization is at an early stage and population densities are low[34]. Moreover, any comparison between Mosquito Alert and ovitrap sensitivity must take into account that the calculation of the former treats the early warning system as having been deployed country-wide. If we also treat ovitraps as a country-wide early warning system, rather than considering only those municipalities in which they were deployed, we can show that its sensitivity is no more than double that of Mosquito Alert but the results depend on a variety of assumptions. For example, if we compare detections from each method across all of Spain, we estimate that ovitraps have at best the same sensitivity as Mosquito Alert. Excluding the known invasion area from that calculation, ovitraps are no more than twice as sensitive as Mosquito Alert and limiting the analysis to the set of municipalities falling within 30 km of the edge of the known invasion area (still excluding the known invasion area itself), ovitrap sensitivity is no more than 1.35 times that of Mosquito Alert. The choice of calculation depends on how one views the sensitivity question: excluding the known invasion area would

seem to be more consistent with the notion of early warning. In all cases, the ratios drop if we increase the estimated specificity of Mosquito Alert (see Methods, Eq. (1)).

**Human–Mosquito encounter probabilities**. Another advantage of the citizen science approach is its ability to provide information about spatio-temporal variation, at both small and large scales, in the probability of humans and mosquitoes coming into contact—a key issue for understanding disease transmission patterns and risks. We use Bayesian multilevel logistic regression to estimate what we term 'biweekly alert probability', the probability of at least one reliable tiger mosquito report being sent through Mosquito Alert from a given geographic cell of ~20 km$^2$ during a 2-week period, conditional on sampling effort (see Methods).

Controlling for sampling effort is crucial to making sense of the reporting data, and sampling effort is itself modelled as a function of time elapsed since the participant downloaded the app as well as intrinsic participant motivation (modelled as random intercepts). Our sampling effort model (Fig. 4) suggests that reporting is most likely immediately after the participant registers, and that reporting propensity decreases with time. A plausible explanation is that over time participants forget that the app is installed. Most users had short participation times (median and mean participation time are 12 and 46 days) but some remained in the study for long periods (25% remains in for

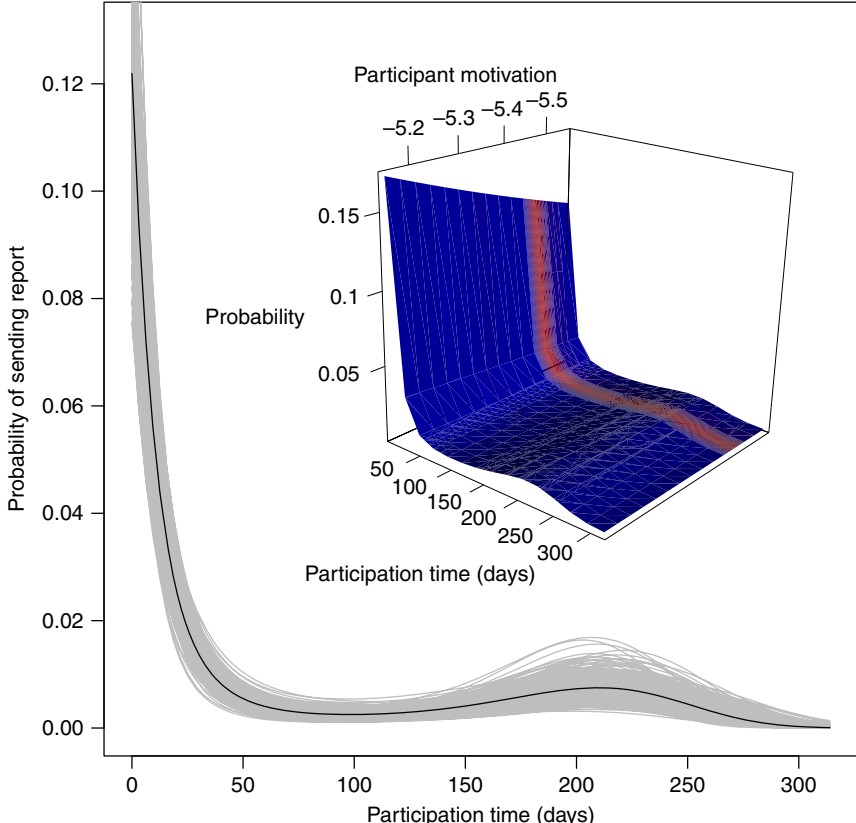

**Fig. 4** Reporting propensity. Probability of participant sending at least one report during a 24-h period as a function of participation time and participant's intrinsic motivation (modelled as random intercept at participant level). Main: values for the mean random intercept. Each grey curve is based on a random draw from the posterior distributions of the participation time parameters; black is based on the mean of the posteriors. Inset: prediction surface with participant motivation (random intercepts) shown on x axis as continuous variable in observed range; surface colour represents density of participants at each random intercept value (red = higher; blue = lower)

at least 57 days). Of these long-term participants, some continued to send in reports while others did not (see Supplementary Fig. 3 for more information on participation times and withdrawal rates, and Supplementary Table 1 for comparison of different reporting propensity models). The reporting propensity model predictions are used to estimate biweekly alert probabilities, which we treat as a proxy for human–mosquito encounter probability.

We make estimates for every day of the mosquito season across Spain. Figure 5 shows mean September biweekly alert probabilities in municipalities across eastern Spain; daily estimates can be viewed in Supplementary Movies 1 and 2. Parameter estimates for all models are shown in Table 2, and described in detail in the Methods section.

To validate our results, we compare them with contemporaneous information about tiger mosquito egg presence from traditional ovitrap sampling in those municipalities in which ovitraps were deployed. Figure 6 shows receiver operating characteristic (ROC) curves for the Mosquito Alert biweekly alert probability predictions at the municipality level, using observed ovitrap egg presence as well as modelled ovitrap egg presence as 'ground truth' (treating the model predictions of >0.5 as positives). These curves provide a way to evaluate the performance of a classifier, with greater area under the curve (AUC) indicating better performance[35, 36]. Importantly, the ROC curve is insensitive to changes in class distribution (i.e., the proportion of positives in the population)[35]. The comparison against the modelled ovitrap results gives an AUC of 0.85 and the comparison against the raw oviposition data gives an AUC of 0.78. In both cases, the areas are well above the value of 0.5

that would result from random guessing. Using the modelled ovitrap results as 'ground truth' gives a better result, but the comparison with the actual observed data is also striking in that the Mosquito Alert model is predicting actual ovitrap egg presence over a 2-week period. If we pick the threshold that gives the highest sum of sensitivity and specificity, we find that the Mosquito Alert model can predict 84% of the egg presence (sensitivity) and 75% of the egg absence (specificity) predicted by the ovitrap model; it can predict 65% of the actual municipality egg presence and 85% of the actual municipality egg absence observed in the ovitraps (Fig. 6). We can see this comparison directly in Fig. 7, which plots all of the daily municipal Mosquito Alert predictions for which there was also ovitrap data in 2015, indicating the ovitrap model prediction (y axis) as well as the observed ovitrap result (colour) and the Mosquito Alert sampling effort (size).

## Discussion

Overall, the results suggest the potential for citizen science to outperform traditional methods in many respects. With its relatively low cost centred on non-recurring investments, citizen science is inherently more scalable than traditional tools (Fig. 8). This makes it possible for citizen science to greatly expand surveillance areas even in the face of shrinking budgets. The ease with which Mosquito Alert has been able to expand across jurisdictions can be seen in its detections far beyond the known invasion front where traditional ovitrap surveillance is usually deployed (Fig. 1)[18, 32, 33]. These reports also rapidly connect apparently isolated invasion areas (Fig. 1). The result is that

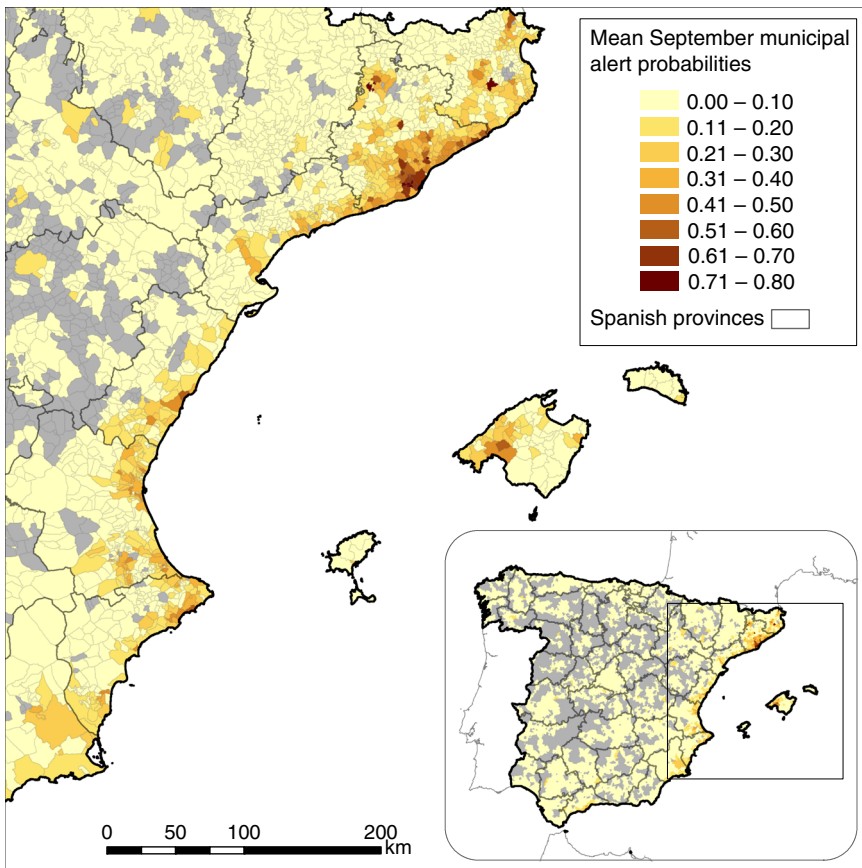

**Fig. 5** Mean September municipal alert probabilities in Spain (Canary Islands, Ceuta and Melilla excluded), based on 2014–2015 Mosquito Alert reliable reports. Grey municipalities were not sampled by Mosquito Alert participants during this period. Boundary data from Spanish National Geographic Institute[53]. © Instituto Geográfico Nacional

Mosquito Alert facilitates early warning surveillance in new regions and control in the areas already colonized.

Detection patterns were complex. Mosquito Alert failed to detect tiger mosquitoes in some municipalities that had positive ovitraps and vice versa. This illustrates the difficulty of finding mosquitoes (and other invasive species) at the edges of the invasion fronts or in other areas undergoing early colonization stages, and the general problem of tracking species that exhibit stratified dispersal[22]. The best solution would appear to be the combination of citizen science with traditional methods, allowing the two approaches to complement one another. We are now working to better integrate citizen science with traditional mosquito surveillance methods, using unified sampling procedures and management plans. Citizen scientists will play increasingly important roles in the process and will be rewarded with near-real-time information about surveillance and control actions in their vicinity. Our hope is that this will also increase public understanding about disease vector mosquitoes, and encourage greater public participation in management efforts, particularly in removing breeding sites from areas that official control services cannot access (like private property). The combination of rich and widespread citizen science information and public involvement with traditional field surveillance and management practices can provide a powerful set of tools for fighting disease vector mosquitoes.

In addition to detecting the invasion front, it is crucial to study the spatial variation in human–mosquito interaction within the known range. This is particularly true in the case of disease vectors, given the direct link to disease risk and transmission rates and existing evidence that both vector density and human–vector interaction can vary substantially across small geographic areas, often linked to complex socio-economic conditions[24–27]. This is also an area in which citizen science can provide good solutions to the problems of scale encountered by traditional methods. Here, however, sampling bias must be addressed[22]. As in other citizen science platforms[30, 37, 38], Mosquito Alert participants self-select into the project—likely based on spatially correlated factors, like exposure to information and socio-economic status[39, 40]. Moreover, while participants have access to the data collection tool (the app) whenever they have their phones, their propensity to report a tiger mosquito when they encounter it likely depends on spatially and temporally correlated factors like whether they are at work, driving a car or relaxing in their backyard. This means that the density of Mosquito Alert observations is unlikely, on its own, to provide a good estimate of tiger mosquito density or the probability of someone (not necessarily a participant) encountering a tiger mosquito. To remedy this problem of sampling bias[41], we model sampling effort (Fig. 4).

Accounting in this way for sampling bias, we find that citizen science gives us almost the same information and predictive capabilities as we get from traditional surveillance tools. If we take the ovitrap results (modelled or observed) as 'ground truth,' we can calculate citizen science sensitivity, specificity and accuracy, as well as the more robust measure of AUC. While citizen science performs well under these tests, the more relevant question for epidemiological purposes may be about the computation of the probability of human–mosquito encounters. For this latter purpose, citizen science provides the more direct—and probably more accurate—estimate. Most importantly, the citizen science results cover a vastly larger geographic area than those

**Table 2 Ovitrap (O1–O6) and Mosquito Alert (MA1–MA6) model estimates**

| | O1 | O2 | O3 | O4 | O5 | O6 | MA1 | MA2 | MA3 | MA4 | MA5 | MA6 |
|---|---|---|---|---|---|---|---|---|---|---|---|---|
| *Main effects* | | | | | | | | | | | | |
| Exposure | 0.095 (0.016) | 0.005 (0.016) | 0.007 (0.017) | −0.003 (0.012) | 0.068 (0.017) | | | | | | | |
| Sampling effort | | | | | | | 1.309 (0.077) | 0.008 (0.075) | 0.010 (0.077) | 0.013 (0.079) | 1.561 (0.076) | |
| Day-of-year | 0.004 (0.001) | 0.016 (0.001) | | | | | −0.0002 (0.001) | 0.006 (0.001) | | | | |
| Day-of-year sq. | −0.061 (0.003) | −0.071 (0.003) | | | | | −0.049 (0.002) | −0.054 (0.002) | | | | |
| Day-of-year cu. | | −0.036 (0.002) | | | | | | −0.032 (0.003) | | | | |
| *Random slopes* | | | | | | | | | | | | |
| Day-of-year | | | Y | Y | Y | Y | | | Y | Y | Y | Y |
| Day-of-year sq. | | | | Y | Y | Y | | | | Y | Y | Y |
| Day-of-year cu. | | | | | Y | Y | | | | | Y | Y |
| *Random intercepts* | | | | | | | | | | | | |
| Sampling cell | Y | Y | Y | Y | Y | Y | Y | Y | Y | Y | Y | Y |
| Ovitrap | Y | Y | Y | Y | Y | Y | | | | | | |
| ELPD | −2362 (54) | −2286 (52) | −2317 (54) | −2883 (53) | −2263 (52) | −2270 (52) | −5697 (96) | −5646 (95) | −5712 (98) | −6692 (108) | −5643 (97) | −5909 (98) |
| N | 10,237 | 10,237 | 10,237 | 10,237 | 10,237 | 10,237 | 31,679 | 31,679 | 31,679 | 31,679 | 31,679 | 31,679 |

Outcome variable is log odds of tiger mosquito egg presence (ovitrap models) or log odds of at least one reliable tiger mosquito report (Mosquito Alert models). For main effects, values in parentheses below each estimate are standard errors, calculated as mean absolute deviation of the posterior distribution. For random effects, 'Y' indicates inclusion in the model. ELPD is Expected Log Pointwise Predictive Density, estimated with the Watanabe-Akaike Information Criterion (WAIC), with standard errors in parentheses calculated as standard deviation of the components that are summed to form the ELPD[60]

from ovitraps, and this allows for predictions about the tiger mosquito distribution across Spain (and potentially across the globe) (Fig. 8).

This discussion shows the importance of considering socio-ecological dynamics. Research on disease-vector mosquitoes has identified gaps in our knowledge of the complex socio-ecological systems in which these mosquitoes are embedded[24]. Variation in and interactions between vector prevalence, exposure, reporting propensity and participation in control efforts across socio-economic gradients show that mosquito control hinges as much on an understanding of humans as it does on an understanding of insects[24]. Citizen science often struggles to attract a socio-economically diverse pool of participants[39], and that challenge is clearly vital to ensuring not only the equitable distribution of benefits stemming from participation, but also sound scientific outcomes.

Another important issue that this discussion raises is the ways in which citizen science information is evaluated and employed. Here it is important to consider 'fitness for use'. That concept, with roots in geography, emphasizes that data quality must be evaluated in light of the particular purpose to which the data will be put[42]. It is an idea at the heart of a growing research field that is rapidly improving the quality and quantity of citizen science data[38, 43, 44]. This improvement potential, along with scalability, makes citizen science particularly powerful for illuminating the human role in mosquito dispersal.

The highly networked form of citizen science discussed here shares many characteristics with other new and emerging methods and data sources centred around the internet and mobile communications devices, including the volume and 'velocity' of the data (the latter term describing the potential for real-time analysis and response), as well as the extent to which it depends on changing patterns of human–computer interaction and crowd dynamics[45]. These latter characteristics suggest that special care needs to be taken with the analysis of citizen science data over time to avoid unexpected biases caused by changes in the algorithms behind data collection platforms or the behaviour of

the people and groups interacting with those platforms. Drawing on experiences in the broader 'big data' context, this means building in robustness checks and validation using multiple data sources, and frequently recalibrating models[46, 47]. For example, the observed patterns of participation time and reporting propensity could well change over time and so should be rechecked regularly, with models adjusted accordingly.

This work is the proof-of-concept that citizen science has huge potential for combatting mosquito-borne diseases. Indeed, our experience has led us to expand Mosquito Alert to include not only tiger mosquitoes but also Yellow Fever mosquitoes (*Aedes aegypti*) and to take an increasingly global focus, working with UN Environment, the Woodrow Wilson International Center for Scholars, the European, Australian and American Citizen Science Associations, the growing citizen science community in Southeast Asia, and mosquito-related citizen science projects and research groups around the world to create the Global Mosquito Alert initiative[48]. Citizen science is a highly scalable and affordable complement to traditional surveillance methods for targeted mosquitoes. Not only can it raise public awareness and involvement, it can also provide useful and scientifically sound data across broader spatial areas and temporal periods. Citizen science can make it possible to replace large and expensive ovitrap deployments with more restricted and prioritized expert management in the field. Finally, adding citizen-science-based vector data to existing habitat suitability[49] and disease[50] models holds particular promise for addressing epidemiological risks at key operational scales, from global to local.

## Methods

**Data collection.** Data were collected in Spain from oviposition traps (ovitraps) and the Mosquito Alert citizen science platform during 2014–2015. Ovitraps are small, dark containers in which tiger mosquitoes lay eggs and in which those eggs can be easily detected. A total of 1558 ovitraps were placed in Spain (Fig. 7), in the provinces of Alicante, Almería, Islas Baleares, Cádiz, Castellón, Granada, Guipúzcoa, Huelva, Huesca, Málaga, Murcia, Sevilla, Valencia. The ovitraps were set repeatedly throughout the year, mostly between June and November, and checked for tiger mosquito eggs after ~2 weeks of exposure.

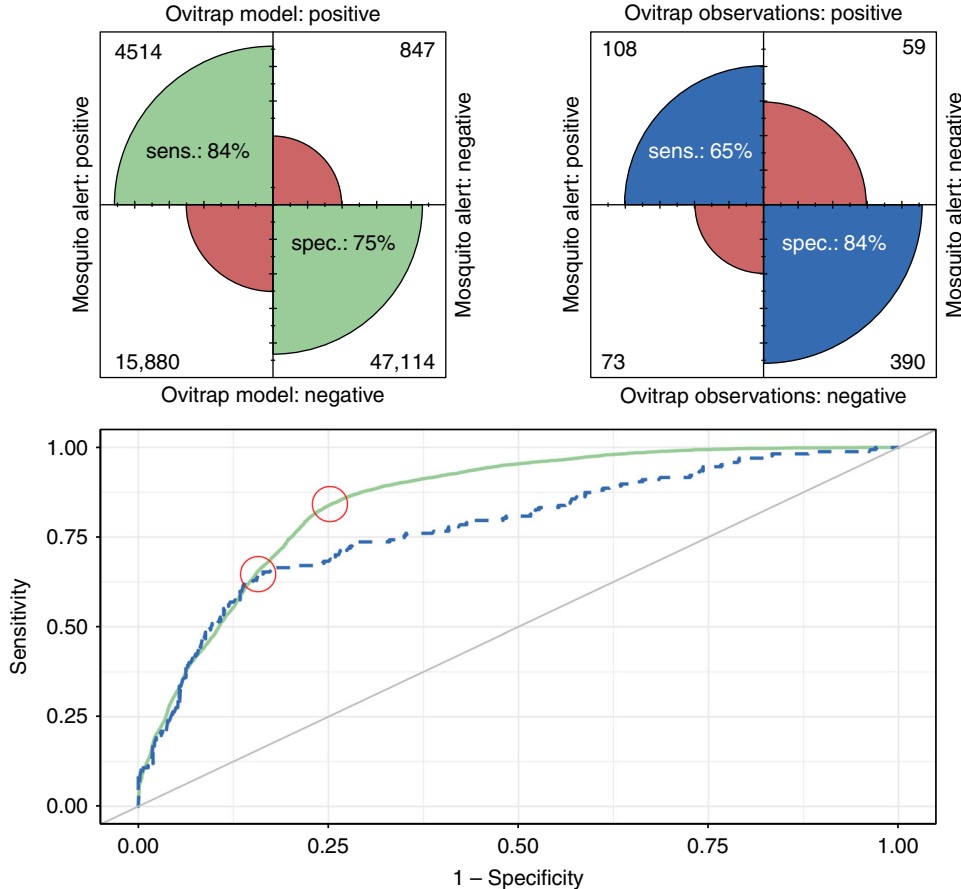

**Fig. 6** Receiver operating characteristic (ROC) curves (bottom) and confusion matrix plots (top) for Mosquito Alert human–mosquito encounter probability model predictions. Bottom: ROC curves show trade-off between sensitivity and specificity of biweek municipality predictions depending on the threshold used for predictions, with the area under the curve (AUC) providing a measure of classifier performance that is insensitive to changes in class distribution. Plot shows Mosquito Alert predictions of biweek municipal-level ovitrap model predictions (solid green line; AUC = 0.85) and observations (dashed blue line; AUC = 0.78). Diagonal line shows AUC = 0.50, which is the theoretical value for random guessing. Top: confusion matrix plots drawn using the thresholds that maximize the sum of sensitivity and specificity for the ROC curve drawn from the ovitrap model (left) and observed ovitraps (right). This threshold is indicated on the ROC curves themselves (bottom plot) with red circles

Mosquito Alert reports have been collected continuously since June 2014 from citizen scientists across the globe, focused mostly in Spain (Fig. 7). The Mosquito Alert mobile phone application helps participants correctly identify tiger mosquitoes and lets them easily create sighting reports (observations) that include geolocation, a brief taxonomic survey (which asked whether (1) the mosquito is it small and black with white stripes, whether (2) it has a white stripe on the head and thorax and whether (3) it has white stripes on the abdomen and legs), optional photographs and optional notes. (In 2016, the system was expanded to include also Yellow Fever mosquitoes, *Aedes aegypti*, given their important role in Zika, dengue and chikungunya transmission in many countries; as a result the taxonomic survey has changed). Reports are subsequently validated by a team of entomologists, based on the attached photographs. The entomological validation scores fall into six categories 'confirmed Tiger', 'probably Tiger', 'unclassifiable', 'probably not Tiger' and 'surely not Tiger'[31]. A report is coded as 'reliable' if it is expert validated under categories 'confirmed' or 'probable', or if the entomological team is unable to give it a score (because it lacks photos or because the photos are not clear) but the participant confirms the observation based on the brief taxonomic survey. Indeed, unsupervised classification of expert-based report scores[51, 52], shows that participant answers to the brief taxonomic survey are among the top five predictors of expert scores (an affirmative triplet, meaning high scoring), together with the number of reports that are close in time and space to the targeted report and the total number of reports sent by the participant[52]. For the early warning analysis, we use the 1976 reports under categories 'confirmed' and 'probable' submitted during 2014–2015 in Spanish territory (excluding Ceuta, Melilla and the Canary Islands). For the human–mosquito encounter analysis, we use the 4767 reliable reports submitted during this time period and from this geographic area. Including survey-based reports (without pictures) allowed us to improve our modelling and statistical significance, and appears qualitatively to produce the same results we would expect from the more accurate but smaller set of 1976 reports under categories 'confirmed' and 'probable'.

Mosquito Alert also collects anonymous information on the geographic distribution of its citizen scientist participants in order to correct for biases caused by uneven sampling effort. Although participants may opt out of this feature, by default the Mosquito Alert Android application uses satellite and network data to estimate the location of the device five times per day at random times between 7:00 am and 10:00 pm (thus excluding the times when tiger mosquitoes are least likely to be active). Before transmitting these locations to the server, the application rounds them down to the nearest 0.05° latitude and longitude, thereby assigning them to sampling cells of ~20 km$^2$ and protecting participants' privacy. No data are collected on the exact location within each sampling cell, or on any other information about the participant. The sampled locations are linked only to randomly generated participant identifiers that are different from the identifiers used for participant reports and other data in order to avoid the risk of re-identification.

**Surveillance economic cost calculations**. Ovitrap costs were estimated from detailed information on the ovitrap program in Murcia province during 2014–2015. That program required ~181,074 Euros to deploy 900 traps at 450 sampling points across the province, taking a total of 13,500 samples over 7 months. For comparison with Mosquito Alert, we aggregate these samples by biweekly period (the approximate duration trap exposure) and sampling cell. The sum of the sampling cell areas across all biweeks sampled is 38,710 km$^2$, which gives an average cost of 4.68 Euros per km$^2$ per biweek or 9.36 Euros per km$^2$ per month. We assume the cost in Spain scales approximately linearly by time and area covered because 97% of the Murcia estimate consists of recurring labour and transportation expenses that depend on distances travelled and areas sampled.

Mosquito Alert costs were calculated from the total project budget for 2014–2015 of 300,000 Euros. For each biweekly period, we take only sampling cells with non-zero sampling effort (meaning that a participant was detected in the cell during that period). The sum of the area of these cells across all biweeks is 487,775 km$^2$, which gives an average cost of 0.62 Euros per km$^2$ per biweek or 1.23 Euros per km$^2$ per month.

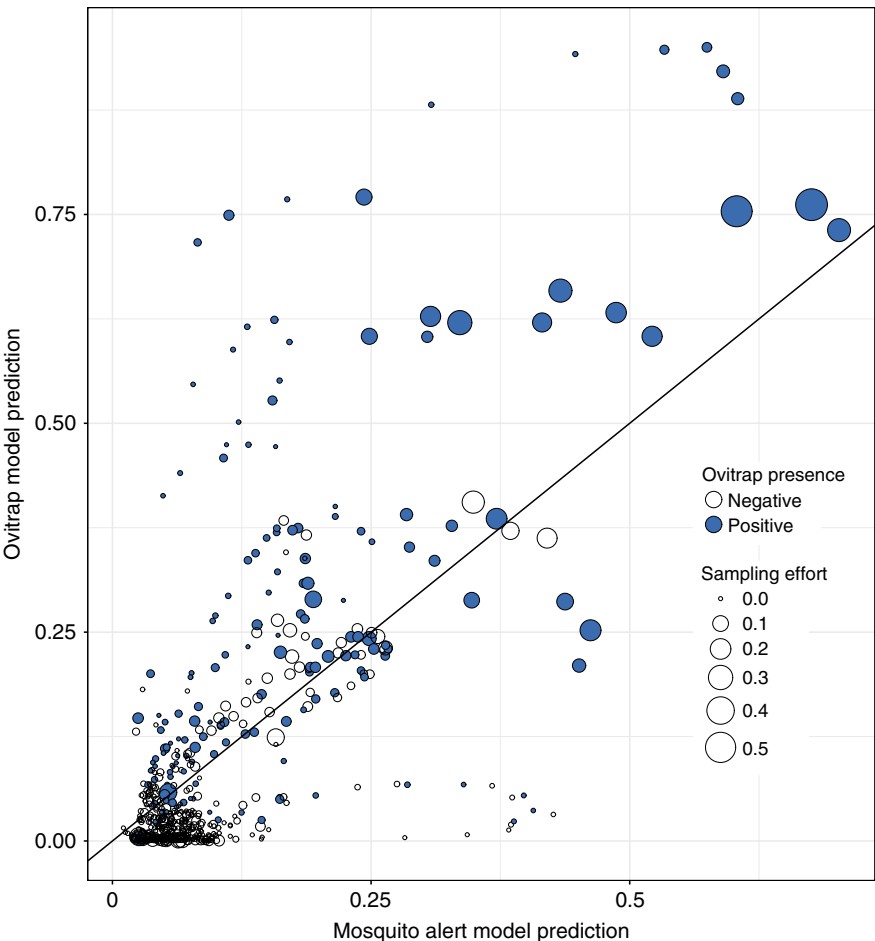

**Fig. 7** Comparison of daily Mosquito Alert with ovitrap predictions and observations for contemporaneously sampled municipalities in 2015. Marker location indicates probability of at least one reliable tiger mosquito report being sent from the municipality during a given 2-week period, as predicted by the Mosquito Alert model (x axis), and of at least one tiger mosquito egg detection in the municipality from a trap left out during the same period, as predicted by the ovitrap model (y axes). Marker colour indicates observed ovitrap results of tiger mosquito egg presence (blue) or absence (white). Reference line indicates equality of ovitrap and Mosquito Alert predictions

**Early warning and surveillance performance analysis**. The performance of Mosquito Alert as an early warning system discussed in the main text is based on a straightforward accounting of the two main surveillance methodologies that have been the basis for the first tiger mosquito detections in each municipality: ovitrap networks and citizen science (Mosquito Alert). For this analysis, we only used the expert validated reports scored by the entomological team as 'probable' or 'definite' tiger mosquitoes. We used Geographic Information Systems (GIS) to compute the number of municipalities, the total area covered, and the distributions of near distances to the invasion front as known at the end of 2013[17]. We factored this information for each of the surveillance methodologies used, distinguishing municipalities where the discoveries have been generated by ovitraps alone, ovitraps alone and in combination with Mosquito Alert, and by Mosquito Alert alone and Mosquito Alert in combination with ovitraps. GIS data sets were obtained from Mosquito Alert data, ovitrap layers[17, 18], from Spanish administrative boundaries[53] and other global baseline layers[54].

To calculate ovitrap early warning system sensitivity, we use the following equation:

$$\frac{\text{TPR}_o}{\text{TPR}_m} = \frac{\text{TP}_o}{\text{TP}_m} = \frac{P_o - N * \text{FPR}_o}{P_m - N * \text{FPR}_m} \tag{1}$$

where $\text{TPR}_m$ and $\text{TPR}_o$ are true positive rates (sensitivities) for Mosquito Alert and ovitraps, respectively, $\text{TP}_m$ and $\text{TP}_o$ are the true positive counts for Mosquito Alert and ovitraps, respectively, $\text{FPR}_m$ and $\text{FPR}_o$ are the false positive rates (1 − specificity) for Mosquito Alert and ovitraps, respective, $P_m$ and $P_o$ are the number of positives detected by Mosquito Alert and ovitraps, and $N$ is the total number of negative municipalities in Spain (i.e., municipalities without tiger mosquito presence).

In 2014–2015 there were 390 municipalities classified as positive by Mosquito Alert ($P_m$), of which 175 were outside the known invasion area. There were also 166 municipalities classified as positive by ovitraps ($P_o$), all of them outside the

known invasion area. We use 0.03 as the false positive rate for Mosquito Alert based on the comparison of 125 municipalities in which ovitraps were placed and failed to detect tiger mosquitoes during 2015 (only four of which were classified as positive by Mosquito Alert) and we assume that the false positive rate for ovitraps is 0. We do not know the total number of negative municipalities in Spain ($N$) but we can put an upper bound on it by subtracting all those classified as positive by ovitraps as of the end of 2015 (538), from the total number of Spanish municipalities (8033, excluding the Canary Islands, Ceuta and Melilla). That gives us 7495 as the maximum value for $N$, which means that the maximum ratio of ovitrap to Mosquito Alert sensitivity (Eq. (1)) is 1.01 if the known invasion area is included, and 2.07 if it is excluded. If we reduce the estimated false positive rate of Mosquito Alert to 0, the ratios fall to 0.43 including the known invasion area and 0.95 excluding it. In other words, as a country-wide early warning system, ovitraps have, at most, twice the sensitivity of Mosquito Alert, but possibly much lower sensitivity.

Including the known invasion area in the calculation may be too far from the notion of early warning and also may bias the result in favour of Mosquito Alert because ovitraps tend not to be placed within that area. On the other hand, excluding the known invasion area while including all other Spanish municipalities may bias the result in the other direction. An alternative comparison might focus only on the edge of the known invasion area. For example, if we consider only the 1398 municipalities that lie within 30 km of the known invasion area (and excluding those in the known invasion area itself), we have 141 classified as positive by Mosquito Alert, 139 classified as positive by ovitraps and a maximum for $N$ of 1259. This gives a maximum ratio of ovitrap to Mosquito Alert sensitivity (Eq. (1)) of 1.35. Reducing the estimated false positive rate for Mosquito Alert to 0 brings the ratio to 0.99.

**Human–Mosquito encounter analysis**. The population distribution analysis is based on modelling ovitrap and Mosquito Alert data. The primary unit of analysis

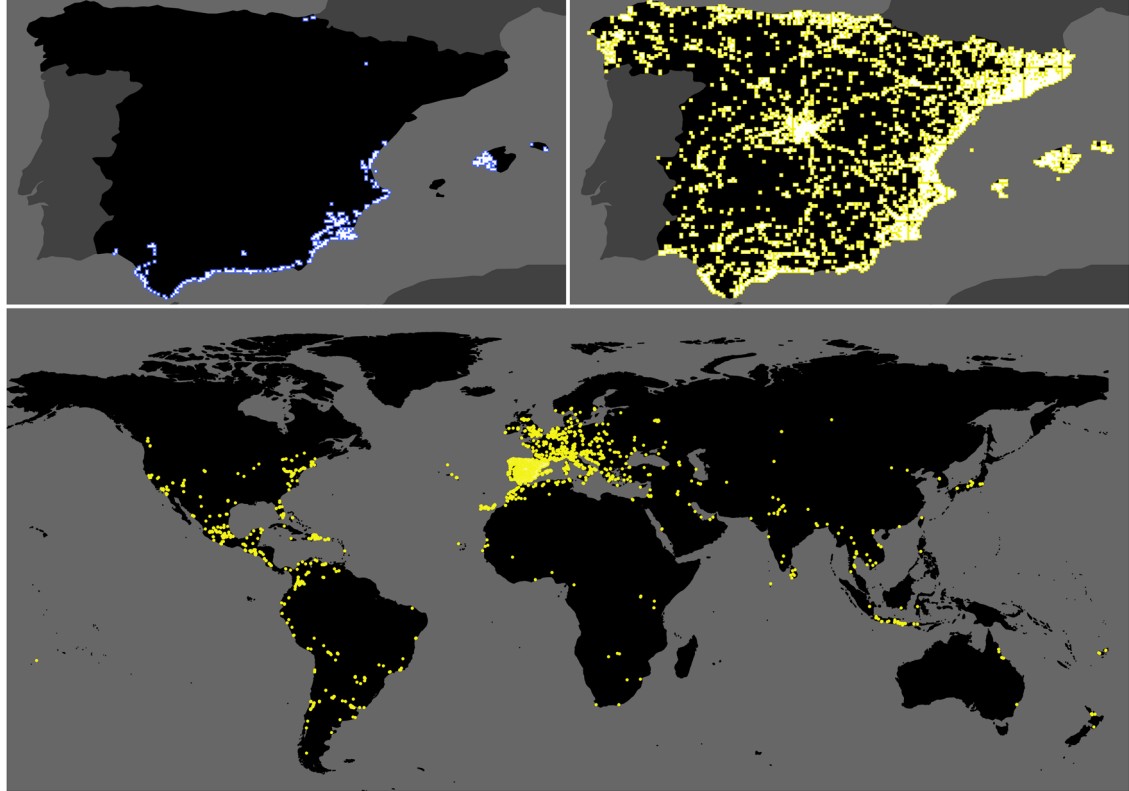

**Fig. 8** Ovitrap and citizen scientist locations. Top left: blue markers show locations of sampling cells containing the 1558 ovitraps used in the analysis, accounting for nearly all ovitraps placed in Spain during 2014–2015. Top right and bottom: yellow markers show locations of sampling cells in which Mosquito Alert participants were randomly sampled in Spain (top right, excluding the Canary Islands, Ceuta and Melilla) and around the globe (bottom) during 2014–2015. Sampling occurred on Android devices five times per day at random times between 7:00 am and 10:00 pm (and only if the participant did not opt out of the background tracking feature). In all cases, white square at centre of marker shows actual sampling cell size, while coloured border is used to ease visualization. Boundary data from Natural Earth (naturalearthdata.com)

for both models is the sampling cell described above, defined by a grid of 0.05° latitude and longitude (~20 km$^2$) and used for masking participant location information. All models, apart from the comparison of ovitrap and Mosquito Alert results, were fit using Hamiltonian Monte Carlo simulations implemented with Stan and R using the rstanarm package[55–58]. Each simulation was run on four parallel chains for 2000 iterations each, with the first 1000 iterations discarded as warmup. Potential scale reduction factor calculations[59] and visual inspection of the chains for each parameter suggested convergence. Goodness of fit and model selection was based on Watanabe-Akaike information criterion (WAIC), calculated using the loo package for R[60, 61]. (WAIC was chosen over the leave-one-out method for computational reasons.) The comparison of ovitrap and Mosquito Alert results was done using ROC curves calculated with the pROC package for R[62].

For the ovitrap data, the probability of tiger mosquito egg presence ($\pi$) at trap $i$ placed for $x$ days in sampling cell $j$ on day-of-year $d$ was estimated in a Bayesian multilevel model with random intercepts for trap ($\alpha_i$) and sampling cell ($\beta_j$) and random slopes at the sampling cell level for day-of-year ($\gamma_j$), day-of-year squared ($\delta_j$) and day-of-year cubed ($\zeta_j$) (in order to capture the nonlinear seasonal cycle). The choice of egg presence (a binary variable) as opposed to number of eggs (a count) as the outcome is based on concerns that the ovitrap egg counts are less reliable and less consistent across ovitraps and study sites. The model treats egg presence as a Bernoulli random variable with probability ($\pi$) and log odds ($\log(\pi/(1 − \pi))$) given mean $\mu$ specified as:

$$\mu_{ijd} = \alpha_i + \beta_j + \gamma_j d + \delta_j d^2 + \zeta_j d^3 + \eta x \qquad (2)$$

The slope and intercept parameters are estimated using weakly informative prior distributions. Each slope parameter is assigned a normal prior distribution with mean of 0 and standard deviation of 100. Estimated parameters were centred and standardized and given weakly informative Cauchy prior distributions with location 0 and scaling factor 2.5. The choice of the Cauchy distribution is based on literature suggesting that it outperforms Gaussian and Laplace alternatives and has other desirable properties[63].

For the Mosquito Alert data, the probability of at least one reliable report ($\pi$) sent from sampling cell $j$ during the 2-week period leading up to day-of-year $d$ given sampling effort $x$ was estimated in a Bayesian multilevel model with random intercepts for sampling cell ($\beta_j$) and random slopes at the sampling cell

level for day-of-year ($\gamma_j$), day-of-year squared ($\delta_j$) and day-of-year cubed ($\zeta_j$) (again, to capture seasonal cycle). The choice of the binary variable of at least one reliable report, as opposed to number of reports, as the outcome is based on concerns over possible autocorrelation in reporting by a given participant or group of participants during short time spans. This choice also has the advantage of making the model easily comparable with the ovitrap model. The model treats the presence of at least one reliable report as a Bernoulli random variable with probability ($\pi$) and log odds ($\log(\pi/(1 − \pi))$) given mean $\mu$ specified as:

$$\mu_{jd} = \beta_j + \gamma_j d + \delta_j d^2 + \zeta_j d^3 + \eta \log(x) \qquad (3)$$

This model is estimated using centred and standardized variables given the same prior distributions as with the ovitrap model in Eq. (2).

While Eqs. (2) and (3) show the ovitrap and Mosquito Alert models discussed in the main text, five alternative models were also tested. These models entailed the following differences: (a) removing the cubic term from the seasonality polynomial, (b) removing the cubic and square terms from the seasonality polynomial, (c) making the seasonality polynomial terms main effects instead of random slopes, (d) making the seasonality polynomial terms main effects instead of random slopes and removing the cubic term and (e) removing the exposure term (in the case of the ovitrap model) and the sampling effort term (in the case of the Mosquito Alert model). Estimates from all models are shown in Table 2.

We prefer the models described in Eqs. (2) and (3) to these alternatives based on theoretical considerations as well as on a comparison of expected log pointwise predictive density (ELPD) estimated as WAIC[60]. From a theoretical perspective, using a third degree polynomial allows a more realistic seasonality curve than the simple parabola forced by a second degree polynomial. Treating this polynomial as a set of random slopes, rather than main effects, allows for seasonal effects to vary by sampling cell and thus should be more reliable over large geographic areas. Finally, exposure (for ovitraps) and sampling effort have logical connections to the probability of egg presence or report transmission, as discussed in the main text. In terms of WAIC, the models in Eqs. (2) and (3) have higher ELPDs than any of the alternatives. For the Mosquito Alert models the difference in ELPDs is greater than twice the ELPD standard error in all cases except for the comparison between the chosen model and the one with the third degree seasonality polynomial as a set of main effects. This could be a basis for choosing the latter model, given its

simplicity. We opted to keep the former, however, based on the theoretical considerations of geographically varying seasonality curves, which should be increasingly important as the project expands. For the ovitrap models, the only ELPD difference that is less than twice the ELPD standard error is that between the chosen model and the model without exposure. This is presumably due to the small variation in exposure times in the data, and exposure was retained in the model on the assumption that this might vary more in future ovitrap data. Model summaries and ELPD estimates are shown in Table 2.

Sampling effort was calculated as the sum of participant-sightings (through the random location sampling of the Android app) in the sampling cell during the two-week period, weighted by each participant's reporting propensity score at the time of sighting. The reporting propensity score is estimated from a model of reporting activity as a function of time elapsed since the participant first downloaded the app and registered with the project (participation time), and intrinsic motivation (Fig. 8). The score is calculated as the marginal probability of a randomly drawn participant ever sending a report, multiplied by the probability of a participant sending at least one report on a given day, conditional on that participant ever sending a report. The conditional probability is modelled as a function of the number of days elapsed since the participant downloaded the app and registered with the project, with a random intercept at the participant level to capture intrinsic motivation (and using only participants who ever sent a report in the model). The outcome variable here is, again, binary, in order to avoid problems of autocorrelation related to the reporting behaviour of participants at short time spans. The model treats the presence of at least one report from participant $i$ sent $x$ days from registration as a Bernoulli random variable with mean ($\pi_i$) and log odds ($\log(\pi_i/(1 - \pi_i))$) given mean $\mu_i$ specified as:

$$\mu_i = \alpha_i + \beta x + \gamma x^2 + \delta x^3 \qquad (4)$$

We estimated this model using the same weakly informative prior distributions as used in the models shown in Eqs. (2) and (3). We also compared this model to alternative specifications using only participation time and participation time squared, using only participation time, and leaving out participation time entirely (see Supplementary Table 1). We selected the model reported here (Eq. 4) based on a combination of theoretical considerations and a comparison of goodness of fit of alternative specifications. From a theoretical perspective, we expected reporting propensity to drop over time, but we found it unlikely that this would be a linear relationship. In terms of model fit, the selected model has higher ELPD than the others, estimated with both the WAIC and leave-one out cross validation (LOO)[60] (Supplementary Table 1).

One complication of fitting this model is that the data are highly skewed towards short participation times: Most users who send a report do so very soon after registering and most drop out of the study after just over a month (median participation time = 12 days), creating a high degree of imbalance in the data set, which complicates estimation[64]. To reduce bias and variance caused by this imbalance, we employed a non-parametric pre-processing technique, dropping the highest 5% and lowest 5% of the participation times and then resampling the data with sampling weights inverse to the proportion of each total participation time (in days) in the remaining data[64]. This resulted in greater balance of participation times, which were then used as the independent variable in the reporting propensity model, improving our parameter estimates.

Both the ovitrap and Mosquito Alert models were used to make daily predictions, for every day of the mosquito season at the sampling cell level. These predicted probabilities were scaled up to the municipality level as follows: for each municipality, 1000 draws were taken from the posterior predictive distributions of the sampling cells falling within the municipality, with the probability of drawing from a given sampling cell's distribution given by the proportion of the municipality's territory covered by that cell. This gives an estimate of the expected value of the probability experienced at a point placed randomly within that municipality.

**Code availability**. The code used in this analysis have been deposited with Zenodo and may be accessed at https://doi.org/10.5281/zenodo.646576[66]. All of the code used for the Mosquito Alert mobile phone applications and server is available at https://github.com/MoveLab.

**Data availability**. The data used in this analysis have been deposited with Zenodo and may be accessed at https://doi.org/10.5281/zenodo.646531[65]. In addition, daily snapshots of the Mosquito Alert data are available at http://doi.org/10.5281/zenodo.597466.

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

## Acknowledgements

We gratefully acknowledge the work of the entire Mosquito Alert team and all the anonymous citizen scientists who have volunteered their time and energy to participate in this project. We also gratefully acknowledge A. Torrell (Departament de Territori i Sostenibilitat, Generalitat de Catalunya) and the support of the Spanish Ministry of Public Health, CCAES (Centro de Coordinación de Alertas y Emergencias Sanitarias) and the Health Department of the Generalitat de Catalunya. J.R.B.P. is funded by the European Union's Horizon 2020 research and innovation programme under Marie Skłodowska-Curie grant agreement no. 657956. The research leading to these results has received funding from the Spanish Ministry of Economy and Competitiveness (MINECO, Plan Estatal I+D+I CGL2013-43139-R) and 'la Caixa' Banking Foundation. Mosquito Alert is currently promoted by 'la Caixa' Banking Foundation. Other contributors during 2014-15 are the Fundación Española de Ciencia y Tecnología (FCT-13-7019), the program RecerCaixa-2013, and the company Lokimica S.A.

## Author contributions

J.R.B.P., A.O. and F.B. created Mosquito Alert, designed the research and carried out the analysis, and R.E. provided entomological expertise at all stages. F.C., J.A.D., J.L., S.D. and M.B. collected and processed the ovitrap data. The citizen scientists who participate in the Mosquito Alert Community carried out the surveillance and reporting of adult tiger mosquitoes. J.R.B.P., A.O. and F.B. wrote the manuscript with input from R.E. and F.C.

## Additional information

**Competing interests:** The authors declare no competing financial interests.

