## [Peer Review file · Nature Communications]

Reviewers' comments:

Reviewer #1 (Remarks to the Author):

The authors describe a novel method for detecting mosquito (*Aedes albopictus*) presence using a reports of citizens through a mobile phone application, called Mosquito Alert. The article provides methodology to estimate the accuracy of Mosquito Alert and in particular its sensitivity and specificity. It also provides estimates of the probabilities of transferring mosquitoes from one Spanish province to another. The methodology is interesting and, more importantly, the results show that Mosquito Alert is a very promising technology that can aid substantially in mosquito detection. Given the current increase in dengue fever prevalence in the world and the recent outbreaks of Zika virus disease and Yellow Fever, the results are very timely. Nevertheless, I find that this article would be much improved by a major rewrite and would benefit a lot with a more standard division of its sections into clearly defined Introduction, Methods, Results and Discussion. It is very difficult right now to distinguish what are the new results and what are part of the methods used (or previous knowledge). I detailed some specific issues below.

Line 54: There is a mismatched quote at the end of "Mosquito Alert"

Lines 70-72: I realized this is a result when I later read the methods. I thought that I was reading the introduction still. You should also remove the term "back-envelope". This is just one example of how this article would improve by following a standard division by Sections.

Line 81: What is meant by the "theoretical invasion front"? Please define prior to its use.

Line 85: You should add as well the number for ovitrap detections on its own and both sources, since even when citizen science detected 107 new municipalities, it missed to detect 99 (detected by ovitrap alone). This is only clear when one goes to the supplement. There should be more discussion of why this is the case, since it is a big issue that both citizen science and ovitrap surveillance have a very low sensitivity. Is it because many municipalities with citizen science do not have ovitrap surveillance, as suggested by Extended Data Figure 1? If so I do not see the importance of the 39%. If ovitrap was present in more municipalities then that number would be much lower. I understand the importance of Mosquito Alert because it is much easier to implement and scale, but this paragraph might be misleading and understood as ovitrap having a low sensitivity. A much more interesting number for me is in how many municipalities where ovitrap and citizen science were present ovitrap did not detect mosquitoes, but there were confirmed (or just even reliable) citizen science reports. Also in the same set of municipalities with ovitrap and citizen science present, in how many did there were no citizen science reports even when ovitrap detected mosquitoes.

Line 86: How many instances?

Line 88: 1.74 is not almost twice. Also is the result statistically significant? Why do "both sources" have a much bigger ratio? Looking at the Extended Data Figure 1, it seems like this is merely a consequence of the fact that ovitraps are set around the coast where most municipalities already had previous detections whereas citizens are evenly distributed around the country.

Line 141: Figure 4b is missing. Is the reference to Figure 4?

Line 153: The Methods Section is almost two thirds of the article. That is very long. A big part of it is because some of the results and discussion are mixed in this Section, but there is also a lot of methods, e.g., details of the statistics, that could be moved to a supplement to the article.

Line 172: What are the three taxonomic questions?

Line 173-177: Is that a result? In any case it shouldn't be on the methods.

Line 180: Why are the expert validated results excluded from the population distribution analysis?

Line 194: This subsection should only state how you calculate the costs not how much the result is.

Lines 222-249 and 310-356: This is mixed results with discussion and methods, it is hard to understand what are actual results.

Line 265-266: The seasonal cycle is captured by including d alone, and non-linearity only needs d^2 , so this justification is not enough. Did you also compare your current model with nested versions using δ_j and/or ζ_j equal to 0?

Line 282: Similar as above.

Line 262: Are there different types of traps that you are varying in i ? How different are they? You should perhaps add this explanation to the methods section. Did you compare with a nested model with $\alpha_i=0$?

Lines 269 and 287: I am confused here, isn't "egg presence" and "at least one reliable report" what are treated as a Bernoulli with parameter π ? If the log odds is Bernoulli as stated its mean μ would be bounded between 0 and 1. So what distribution are you using for the log odds?

Line 308: No justification is provided for using non-linear terms.

Line 311: The issue of short participation time should be discussed at length. What will happen when new user registration decreases at a location? This should be a major concern for Mosquito Alert. In fact, it would be helpful if the article provides data on registration besides the median participation time.

Figure 1: I would suggest to change the light blue color for gray and to drop the current coloring of the provinces. It is enough to have the limits of the provinces to distinguish which ones had previous detections. It is difficult to distinguish light blue from dark blue in isolation, for example in Ibiza. Much more important here is to know what municipalities had both citizen science and ovitrap surveillance. Perhaps an extra figure of just those municipalities colored is justified.

Figure 3: Why does the scale go in increments of 10, except for the last one (0.41-1)?

Extended Data Table 1: Is the new area discovered the sum of the sizes of the municipalities. If so then it shouldn't be called new area discovered as you cannot guarantee that all the area on the newly discovered municipality has mosquitoes. "Mean near distance to invasion front" needs to be defined. Ratio should have 95% CI.

Reference 7: Title of article is missing. Is tigatrapp the same as Mosquito Alert?

SI Video 1: I can only see a fix image. Video 2 works properly. It is a very nice video, by the way. I have the same question as for Figure 3.

Reviewer #2 (Remarks to the Author):

I have reviewed the manuscript by Dr. Palmer and others. This paper presents a large-scale citizen science program that is currently documenting incidence of human-Tiger mosquito contacts in Europe. The Mosquito Alert program is well-documented here and in web resources and the authors compare these citizen-science data with more standardized ovitrap data throughout the region. I feel that the topic and presentation warrant publication, although there are several points I would like to see clarified in the manuscript.

There is a lot going on in this manuscript. The main component is the validation of the Mosquito Alert data and documentation of how effort and reliability were handled. This is generally pretty clear but I did find it difficult to keep track of which statistics/summaries were based on validated reports versus any report. For instance, in Line 84-86, how many of the 'new' municipalities were validated reports? It would be good to see some clearer assessment of uncertainty around summary statistics and discussion specifically about how the uncertainty factors in to spatial detects (i.e., un-validated reports in a municipality with validated reports versus unvalidated 'new' municipalities?). This is related to similar comments about percent overlap on lines 116-120. What are the spatial patterns within the overlap? How often does overlap include/confirm new detections?

The source-sink component of the paper is harder to follow. There are some pretty strong definitions of source-sink dynamics in metapopulation theory but that doesn't seem to be how the authors are using it here. Still, I'm not sure from the text/methods how they are defining source or sink sites. I think this is more about dispersal potential and I'd recommend reworking the text around clarified definitions. The idea is interesting - but for example, I wasn't clear on how to interpret (contextualize/assess uncertainty) the conclusion that Madrid is a 'net sink' for mosquitoes when presence there hasn't been confirmed.

Minor:

Line 70. I'd remove the term 'back-of-the-envelope' and instead refer to it as an estimate.

Line 123. The use of 'fitness' in this sentence is awkward. Perhaps the 'value' of citizen science data?

Line 222. Perhaps replace 'world' with global.

Line 242. Quantify 'Most'.

Line 276. Why were half-Cauchy priors chosen (e.g., versus uniform or inverse gamma)?

Lines 313-315. I don't understand the purpose or method of resampling data or the weights inverse to participation time. Is this just a method to normalize the sampling effort variable?

Detailed Responses to Reviewers

General Points:

We appreciate very much the reviewers' helpful suggestions. We have made a major revision to the manuscript, expanding the main text, improving the structure and organization, clarifying confusing points, and adding new analysis and explanation. Among the more important changes, we have:

- Moved into the main manuscript what had been Extended Data Figures 1-3.
- Improved our analysis of citizen science and ovitraps as early warning systems, including a more thorough analysis of differences in distance to invasion front (see new Figure 3), as well as estimates of sensitivity and specificity.
- Expanded our comparison of citizen science and ovitraps in terms of estimating tiger mosquito prevalence and human-mosquito interaction. This includes new Figures 6 and 7.
- Moved our analysis of inter-province tiger mosquito flows to another paper because that analysis relies on a link between tiger mosquito prevalence (i.e. the *Mosquito Alert* probability computed in this manuscript) and the probability of a mosquito being transported in a car. Such a link is made explicit in a paper under review with *Scientific Reports*, in which we include the results of a large road-side sampling study. In that study we detected mosquitoes in cars, placing us in a better position to properly model mosquito fluxes. We believe that moving this part of the analysis to the other paper helps to focus the present article and also avoids having a preliminary estimate here that would then be modified and improved in another paper running almost in parallel.

Reviewer #1:

1. The authors describe a novel method for detecting mosquito (*Aedes albopictus*) presence using a reports of citizens through a mobile phone application, called Mosquito Alert. The article provides methodology to estimate the accuracy of Mosquito Alert and in particular its sensitivity and specificity. It also provides estimates of the probabilities of transferring mosquitoes from one Spanish province to another. The methodology is interesting and, more importantly, the results show that Mosquito Alert is a very promising technology that can aid substantially in mosquito detection. Given the current increase in dengue fever prevalence in the world and the recent outbreaks of Zika virus disease and Yellow Fever, the results are very timely. Nevertheless, I find

that this article would be much improved by a major rewrite and would benefit a lot with a more standard division of its sections into clearly defined Introduction, Methods, Results and Discussion. It is very difficult right now to distinguish what are the new results and what are part of the methods used (or previous knowledge). I detailed some specific issues below.

Thank you very much for this suggestion. We have taken the opportunity to do a major revision, expanding the manuscript into the standard division of sections, as suggested. We have also better clarified and separated the methods from the other sections.

2. Line 54: There is a mismatched quote at the end of "Mosquito Alert"

Fixed.

3. Lines 70-72: I realized this is a result when I later read the methods. I thought that I was reading the introduction still. You should also remove the term "back-envelope". This is just one example of how this article would improve by following a standard division by Sections.

We have removed the term “back-of-the envelope”, and we have elaborated on the cost calculations in the first subsection of the Results section (Lines 111-19).

4. Line 81: What is meant by the "theoretical invasion front"? Please define prior to its use.

We have changed our wording and explained better that by “theoretical” invasion front we were making reference to the “known” invasion front. And we have improved our explanation about what this is: The border of the area in which tiger mosquito presence is already known. The new text (lines 122-27) reads:

We observe that Mosquito Alert accounts for first detections far beyond the known invasion area, to which traditional surveillance methods are usually limited (Fig. 1). The known invasion area is comprised of the municipalities in which the mosquito has already been detected, and ovitraps are usually deployed inside this area or in municipalities contiguous with its edges (the invasion front). The Mosquito Alert detections of tiger mosquitoes far beyond this area demonstrate that the invasion process can occur in jumps.

4. Line 85: You should add as well the number for ovitrap detections on its own and both sources, since even when citizen science detected 107 new municipalities, it missed to detect 99 (detected by ovitrap alone). This is only clear when one goes to the supplement.

We had previously been using a much more concise manuscript structure. Now that we have expanded it, we have included this comparison as well as additional discussion of sensitivity and specificity of *Mosquito Alert* and ovitraps as early warning systems. The relevant text in the new version (129-96) reads:

The comparison between Mosquito Alert and ovitraps is complicated somewhat by the fact that Mosquito Alert detections in new municipalities triggered ovitrap deployment in some of these municipalities^{1,2}. We know that this occurred in at least 11 cases^{1,2}, covering the regions of Andalusia (5)¹, Murcia (4), Aragon (1) 27, and Catalonia (1), and we suspect others in Valencia and the Balearic Islands. Quantifying this is difficult because new Mosquito Alert detections are made public and also reported to the Centre for the Coordination of Health Warnings and Emergencies (CCAES), which in turn notifies the regional public health actors. The latter must investigate tiger mosquito reports in the field in order to officially confirm the species' presence, and this is usually done by deploying ovitraps. If presence is confirmed, mosquito public health protocols and management actions are activated, and the new discovery is often reported in scientific publications^{1,2}, but the source of the initial information that led the regional public health actors to deploy ovitraps is not always clear. Figure 2 illustrates how this process played out in 2015 in the south of Spain, after Mosquito Alert first detected tiger mosquitoes there the previous year¹. In Andalusia, ovitraps were deployed in 5 municipalities from which reliable Mosquito Alert reports had originated, but also in many other municipalities along the coastline (Fig. 2). In 4 out of these 5 municipalities tiger mosquito presence was confirmed in the field by ovitraps.. Given this connection between Mosquito Alert reports and ovitrap deployment, we distinguish, in the following analysis, between the non-overlapping sets of municipalities in which ovitraps or Mosquito Alert observations are alone responsible for the detection and the overlapping sets of total municipalities for which each source is credited.

Of 274 Spanish municipalities in which tiger mosquitoes were detected for the first time in 2014-15, Mosquito Alert is alone credited with detections in 108 (39%). In total, Mosquito Alert is credited with detections in 175 municipalities (64%). In contrast, ovitraps are alone credited with 99 municipalities (36%), with a total of 166 (61%). The municipalities for which Mosquito Alert is alone credited and those for which ovitraps are alone credited cover approximately the same area: 5,761 and 5,787 km², respectively, as do the totals for each source: 10,948 and 10,975 km².

The known invasion front is contiguous with only 19% of the municipalities for which Mosquito Alert is alone credited and 25% of the total Mosquito Alert municipalities. In contrast, the front is contiguous with 60% of the municipalities for which ovitraps are alone credited and 50% of the total

ovitraps municipalities. A simple Chi-squared test of the equality of these proportions against the alternative that the ovitraps proportion is greater yields p-values well below 0.01, regardless of whether we use the non-overlapping or the overlapping sets of distances. Thus, the observed differences are very unlikely to be the result of random variation alone.

The median and mean distances to the invasion front of the municipalities for which Mosquito Alert is alone credited are 17 km and 37 km, while those of the municipalities for which ovitraps are alone credited are 0 km (i.e. contiguous) and 20 km. If we include the overlapping municipalities in these calculations, the Mosquito Alert median drops only slightly, to 16 km, and its mean rises to 43 km, while the ovitraps median remains less than 0.5 km and its mean rises to 34 km. The difference between these distributions is obvious in Fig. 3 and strongly supported by Mann-Whitney tests, which yield p-values well below 0.01 regardless of whether overlapping municipalities are included. The geographic scale of the Mosquito Alert detections is especially apparent in the new detections located far to the south and west of the known invasion front (Fig. 1)^{2,3} and the differences would be even more pronounced had Mosquito Alert detections not triggered overlapping ovitraps detections.

We are confident that nearly all of the Mosquito Alert detections are true positives based on a comparison with municipalities in which ovitraps were deployed contemporaneously: Of 125 municipalities in which ovitraps were deployed and failed to detect tiger mosquitoes in 2015, only 4 were classified as positive by Mosquito Alert. This gives a specificity (true negative rate) of 97%, assuming that ovitraps have perfect sensitivity, and higher if the ovitraps failed to detect true positives in any of these municipalities. This high specificity makes sense, given that the Mosquito Alert early warning system relies only on citizen scientists' observations that include photographs that are subsequently validated by experts (see Methods).

We also use the comparison with ovitraps to estimate sensitivity. Of 112 municipalities in which ovitraps were deployed and detected tiger mosquitoes in 2015, 53 were classified as positive by Mosquito Alert, giving a sensitivity (true positive rate) of 47%. This figure must be interpreted in light of Mosquito Alert's large geographic coverage and the general problem of surveillance sensitivity at the edges of invasion fronts, where colonization is at an early stage and population densities are low (pers. comm. A.Torrell. Dept. Territori i Sostenibilitat, Generalitat de Catalunya). Moreover, any comparison between Mosquito Alert and ovitraps sensitivity must take into account that the calculation of the former treats the early warning system as having been deployed country-wide. If we also treat ovitraps as a country-wide early warning system, rather than considering only those municipalities in which

they were deployed, we can show that its sensitivity is no higher than Mosquito Alert's (see Methods, Equation 1).

There should be more discussion of why this is the case, since it is a big issue that both citizen science and ovitrap surveillance have a very low sensitivity.

This is an important point. As explained above, we estimate *Mosquito Alert* sensitivity as an early warning system in 2015 to have been about 47% and we estimate the sensitivity of ovitraps as an early warning system to have been no greater than that (and possibly less). These figures should be understood in light of the fact that surveillance systems generally have low sensitivities. There are a variety of reasons for this, but chief among them is the fact that these systems are tasked with detecting invasive species outside areas in which they are established -- and thus where they initially have low population densities. What we find important about *Mosquito Alert* is that its sensitivity is at least as good as the ovitraps and that this is complemented by its broad geographic range. We also expect that the *Mosquito Alert* sensitivity will increase over time as more people volunteer as citizen scientists. (In many ways, this sensitivity question is linked to the cost question; Ovitrap early warning system sensitivity could be increased by deploying more traps, but that does not appear to be a realistic option given the costs involved. *Mosquito Alert*, in contrast, will continue to expand from the initial non-recurring investments we have made in it.)

Is it because many municipalities with citizen science do not have ovitrap surveillance, as suggested by Extended Data Figure 1? If so I do not see the importance of the 39%. If ovitrap was present in more municipalities then that number would be much lower. I understand the importance of Mosquito Alert because it is much easier to implement and scale, but this paragraph might be misleading and understood as ovitrap having a low sensitivity.

We have substantially revised this text, and the key points we try to make are that ovitrap sensitivity as an early warning system should be thought of, like *Mosquito Alert* sensitivity, in terms of its detection of tiger mosquitoes throughout Spain, not just in the municipalities in which it was deployed. In these terms we show that ovitraps had no higher sensitivity in 2015 than *Mosquito Alert*.

A much more interesting number for me is in how many municipalities where ovitrap and citizen science were present ovitrap did not detect mosquitoes, but there were confirmed (or just even reliable) citizen science reports. Also in the same set of municipalities with ovitrap and citizen science present, in how many did there were no citizen science reports even when ovitrap detected mosquitoes.

This is an excellent point. Of the 125 municipalities in which ovitraps were deployed and failed to detect tiger mosquitoes in 2015, *Mosquito Alert* detected tiger mosquitoes in 4. We use this as an estimate of *Mosquito Alert* specificity (assuming the ovitrap results were true negatives and the *Mosquito Alert* results, thus false positives), but we suspect that the

Mosquito Alert results are in fact true positives, since ovitraps often fail to detect tiger mosquitoes even where they are present – and especially in areas that are undergoing colonization and thus have low population densities. We have now explained this at lines 178-85 of the manuscript (see also the text quoted in our answer to question 4 above).

5. Line 86: How many instances?

As we now explain at lines 129-48 (see text quoted above in answer to question 4), we know confirm that this occurred in at least 11 instances, covering the regions of Andalucía (5)¹, Murcia (4), Aragón (1) 27, and Catalonia (1). We suspect additional instances in Valencia and the Balearic Islands. Quantifying this is , however, because tiger mosquito surveillance and control is done at many administrative levels and from different sectors (municipalities, provinces, regions, central government, universities, private companies, etc.) without a common working framework.

6. Line 88: 1.74 is not almost twice. Also is the result statistically significant?

In our revised version (quoted above) we no longer make this multiplicative comparison but instead compare the distributions of distances and their means and medians in much more detail. We also include tests of statistical significance (which show that the differences are indeed significant).

Why do "both sources" have a much bigger ratio? Looking at the Extended Data Figure 1, it seems like this is merely a consequence of the fact that ovitraps are set around the coast where most municipalities already had previous detections whereas citizens are evenly distributed around the country.

As noted above, we no longer use the ratio of means as a comparison, but instead focus in more detail on the distributions as well as differences in means and medians. In addition, instead of using the "both" category on its own (which is problematic, as the reviewer suggests), we use the category of *Mosquito Alert* alone and in combination with ovitraps, and the category of ovitraps alone and in combination with *Mosquito Alert*. (We then also compare each alone.) The ovitraps are definitely concentrated along the coast, based on the fact that this is where the known invasion area lies. The citizen scientists, in contrast, are more evenly distributed around the country (or at least evenly relative to population density).

7. Line 141: Figure 4b is missing. Is the reference to Figure 4?

Thank you - fixed. It should have been Figure 4. In fact, we have now removed this figure from the revised text in order to better focus the paper.

8. Line 153: The Methods Section is almost two thirds of the article. That is very long. A big part of it is because some of the results and discussion are mixed in this Section, but there is also a lot of methods, e.g., details of the statistics, that could be moved to a supplement to the article.

Based on this suggestion, we have moved part of the Methods section to the Results and Discussion sections and we have revised everything to provide clearer structure. The methods section is now approximately 2700 words, while the main text is approximately 4200 words. We think the manuscript is now much better balanced this way.

9. Line 172: What are the three taxonomic questions?

This sentence makes reference to a brief taxonomic survey that participants are obliged to answer when they submit an observation via the Mosquito Alert app. During the time period described in the article, the questions asked of participants about their mosquito were: (1) Is it small and black with white stripes?, (2) Does it have a white stripe on the head and thorax?, and (3) Does it have white stripes on the abdomen and legs? These questions have changed in more recent versions of the app as we are now targeting both tiger mosquitoes and yellow fever mosquitoes and thus we ask for more information to identify and distinguish these two species. We have now added this information to the Methods section (lines 342-44).

10. Line 173-177: Is that a result? In any case it shouldn't be on the methods.

This is not a result from the work being reported here. This is just a methodological aspect where we explain what report classes are considered “reliable” reports. The subset of reports classified as “reliable” are the ones used to compare with ovitraps in this manuscript (explained in Methods section).

11. Line 180: Why are the expert validated results excluded from the population distribution analysis?

The expert validated results are not excluded from the “population distribution” analysis (which we are now calling “human-mosquito encounter” analysis). Our text was not sufficiently clear about this so we have revised it to better explain that this part of the analysis was based on 4767 reports that we classify as reliable. This category includes all reports that the expert validators scored as “confirmed” or “probable” tiger mosquitoes (i.e. all reports used for the early warning analysis), but in addition, this category includes reports that did not include photographs, and thus were not validated by the experts, but that had answers to the taxonomic survey that were consistent with the observation being a tiger mosquito. This extended set of reports accounted for better statistical inferences in our model, although we also tested that our main modeling results are independent of the set used.

Thus, the “human-mosquito encounter” analysis uses a set of reports that includes all of those used in the “early warning analysis”, but that is broader. The early warning analysis, in contrast, uses the smaller set that the expert validators scored as “confirmed” or “probable” tiger mosquitoes (a total of 1976 reports). We have explained all of this at lines 339-64.

12. Line 194: This subsection should only state how you calculate the costs not how much the result is.

Our discussion of the cost calculation in the Methods section is now on lines 379-88. We have tried to better separate methods from results in general but in this case we think it is important to include some of the actual costs in the methods in order to better explain the calculations. The Results section reports that *Mosquito Alert* cost about 1.23 Euros per km² per month whereas ovitraps cost about 9.36 Euros per km² per month (lines 111-19) but to see how we get to these numbers it is helpful to have some of the starting figures and intermediate calculations that we report in the Methods section.

13. Lines 222-249 and 310-356: This is mixed results with discussion and methods, it is hard to understand what are actual results.

We appreciate this comment and have reorganized the manuscript as explained above. In terms of these particular lines, they are now located as follows:

The discussion in lines 222-249 of the previous draft (Methods) is now in lines 129-48 (Results).

The discussion in lines 321-23 of the previous draft (Methods) is now in lines 207-11 (Results).

The discussion in lines 347-56 of the previous draft (Methods) has now been revised based on new analysis and is now in lines 224-44 (Results) and 288-97 (Discussion).

14. Line 265-266: The seasonal cycle is captured by including d alone, and non-linearity only needs d^2 , so this justification is not enough. Did you also compare your current model with nested versions using δ_j and/or ζ_j equal to 0?

The model that we present in the main results and discussion is one selected from a set of models based on goodness of fit and theoretical considerations. We tested a variety of alternative models, including the ones proposed here, and we now include much more information about these alternative models and our basis for selecting among them in lines 436-40 474-501 and in Table 2 of the main manuscript. In terms of the seasonal cycle, adding only the second degree polynomial forces the cycle to follow a parabolic form, whereas the cubic terms allows for additional complexity -- which we think is justified from a theoretical perspective and also achieves a model with better fit based on a comparison of Expected Log Pointwise Predictive Density, estimated with the Watanabe-Akaike Information Criterion (WAIC)⁴. Similarly, we tested models in which the seasonality terms were included as main effects rather than random slopes. These also mainly performed worse than the selected model. We also prefer the random slopes version because it allows the model to be fit over larger geographic areas, as we would expect the seasonality curves to be very different depending on latitude and other factors.

15. Line 282: Similar as above.

Same as above.

16. Line 262: Are there different types of traps that you are varying in i ? How different are they? You should perhaps add this explanation to the methods section. Did you compare with a nested model with $\alpha_i=0$?

There are small differences in some of the traps themselves but the main reason for including trap-level random intercepts is to account for differences in local conditions where each trap is placed and the fact that individual traps are checked multiple times across the season. For example, if a given trap may be placed very close to an important breeding site or in some area that happens to be inhospitable to tiger mosquitoes. The trap-level intercepts account for these possibilities.

17. Lines 269 and 287: I am confused here, isn't "egg presence" and "at least one reliable report" what are treated as a Bernoulli with parameter π ? If the log odds is Bernoulli as stated its mean μ would be bounded between 0 and 1. So what distribution are you using for the log odds?

This was our mistake in the way we phrased it. We have now changed the text to: “The model treats egg presence as a Bernoulli random variable with probability (π) and log odds ($\log(\pi/(1 - \pi))$) given mean μ specified as . . .” (for ovitraps; lines 449-50) and “The model treats the presence of at least one reliable report as a Bernoulli random variable with probability (π) and log odds ($\log(\pi/(1 - \pi))$) given mean μ specified as . . .” (for *Mosquito Alert*, lines 468-69).

18. Line 308: No justification is provided for using non-linear terms.

As with our selection of non-linear terms in the ovitrap and *Mosquito Alert* models (see response to point 14 above), our selection of non-linear terms in the reporting propensity model was based on a combination of theoretical considerations and a comparison of goodness of fit of alternative specifications. From a theoretical perspective, we expected reporting propensity to drop over time, but we found it unlikely that this would be a linear relationship. In terms of model fit, the model with both squared and cubed terms of participation time fits better than any of the alternatives based on a comparison of Expected Log Pointwise Predictive Density, estimated with both the Watanabe-Akaike Information Criterion (WAIC) and leave-one out cross validation (LOO)⁴. (For the main models, we make comparisons based only on the WAIC for computational reasons; for the reporting propensity models, which have much smaller data sets, we used both WAIC and LOO.) We have now included a supplemental table showing these model comparisons (Supp. Tab. 1).

19. Line 311: The issue of short participation time should be discussed at length. What will happen when new user registration decreases at a location? This is should be a major concern for Mosquito Alert. In fact, it would be helpful if the article provides data on registration besides the median participation time.

This is an important point that we now discuss further at lines 207-18 and 307-18. We have also added a new Supplemental Figure 3, showing total participation times and participation-time-specific withdrawal rates (analogous to age-specific mortality rates). How well we engage participants and how they interact with the *Mosquito Alert* system are top concerns of ours, which we have also written about in Oltra et al. (2016)⁵. Our sampling effort estimates, based on the background tracking of participant locations and our reporting propensity model, are key to making good estimates of human-mosquito encounter probabilities from these data. These are designed specifically to deal with the possibility of registration decreasing at a given location (or increasing at another). However, it is critical that we continue to update and recalibrate the reporting propensity model regularly because it may change due to changes in the way people interact with the app or the system as a whole. This has been one of the clear lessons of other studies that harness large groups of people through digital devices and networking (for instance Google Flue Trends⁶) and we now note in the Discussion (Lines 307-18) how relevant it is to our project.

20. Figure 1: I would suggest to change the light blue color for gray and to drop the current coloring of the provinces. It is enough to have the limits of the provinces to distinguish which ones had previous detections. It is difficult to distinguish light blue from dark blue in isolation, for example in Ibiza. Much more important here is to know what municipalities had both citizen science and ovitrap surveillance. Perhaps an extra figure of just those municipalities colored is justified.

We agree with the reviewer, so we have changed the light blue to light grey and removed the colouring of the provinces (now indicated only with a black contour line). Overlapping citizen science and ovitrap detections are indicated with a red-pink colour (see legend “both sources”), so we do not think that we need an extra figure to display those. We have tried to clarify the legend, changing “Citizen science” for “Citizen science alone” and “Ovitrap surveillance” for “Ovitrap alone”. Also, we have added a new feature in the map: two red circles that indicate far away areas from the invasion front, that although are marked as “both sources”, the trigger of the discoveries was citizen science (see reviewer comment about Line 88 and near distances from invasion front)^{1,2}.

21. Figure 3: Why does the scale go in increments of 10, except for the last one (0.41-1)?

We did this to visually enhance higher probabilities and make colours easier to differentiate in the map. However, we agree that this was confusing and we have now changed the map scale to an equal interval scale of 0.09, from the minimum to the maximum value (0-0.80). (This is now Figure 5 in the revised version.)

22. Extended Data Table 1: Is the new area discovered the sum of the sizes of the municipalities. If so then it shouldn't be called new area discovered as you cannot guarantee that all the area on the newly discovered municipality has mosquitoes. "Mean near distance to invasion front" needs to be defined. Ratio should have 95% CI.

What we referred to as “new area discovered” was indeed the sum of the areas of the newly-detected municipalities and the reviewer’s suggestion is a much better way to phrase this. This table is now Table 1 (in the main manuscript), and we have changed the title of the column in question to “Area of newly-detected municipalities”. We have also ensured that the wording in the text is clear on this point (lines 154-56).

We have also changed our analysis of distances to invasion front. In the revised version, we show the mean, median, distribution of near distances to the invasion front. The near distances are calculated as the shortest distance between the border of the newly-detected municipality and the border of the known invasion area as of the end of 2013. This is explained at lines 158-76, Figure 3, and Table 1.

23. Reference 7: Title of article is missing. Is tigatrapp the same as Mosquito Alert?

We have added the title to this reference (currently reference 25): “AtrapaelTigre.com: enlisting citizens-scientists in the war on tiger mosquitoes”. The project was originally named *AtrapaelTigre* (“CatchtheTiger”) and the app was named *Tigatrapp*. We changed both names in the beginning of 2016 to *Mosquito Alert* in order to appeal to global audience.

24. SI Video 1: I can only see a fix image. Video 2 works properly. It is a very nice video, by the way. I have the same question as for Figure 3.

Fixed.

Reviewer #2:

I have reviewed the manuscript by Dr. Palmer and others. This paper presents a large-scale citizen science program that is currently documenting incidence of human-Tiger mosquito contacts in Europe. The Mosquito Alert program is well-documented here and in web resources and the authors compare these citizen-science data with more standardized ovitrap data throughout the region. I feel that the topic and presentation warrant publication, although there are several points I would like to see clarified in the manuscript.

1. There is a lot going on in this manuscript. The main component is the validation of the Mosquito Alert data and documentation of how effort and reliability were handled. This is generally pretty clear but I did find it difficult to keep track of which

statistics/summaries were based on validated reports versus any report. For instance, in Line 84-86, how many of the 'new' municipalities were validated reports?

All reports that include photographs are validated by entomological experts and the early warning analysis (which lines 84-86 were part of in the previous draft) relies only on reports that the experts scored as “probable” or “confirmed” tiger mosquitoes. In contrast, our analysis of human-mosquito encounter probability is based on a wider pool of reports, including also those without photographs but with participants’ responses to the taxonomic survey consistent with their find being a tiger mosquito.

Based on this comment and comments from reviewer #1, we have tried to be more clear in our discussion of this issue (e.g., lines 348-64) as well as of the question of validation of *Mosquito Alert* results in the field using ovitraps (e.g., lines 129-49).

2. It would be good to see some clearer assessment of uncertainty around summary statistics and discussion specifically about how the uncertainty factors in to spatial detects (i.e., un-validated reports in a municipality with validated reports versus unvalidated 'new' municipalities?). This is related to similar comments about percent overlap on lines 116-120. What are the spatial patterns within the overlap? How often does overlap include/confirm new detections?

This is an excellent point and we have now addressed these questions in our discussion of the early warning results at lines 151-97.

3. The source-sink component of the paper is harder to follow. There are some pretty strong definitions of source-sink dynamics in metapopulation theory but that doesn't seem to be how the authors are using it here. Still, I'm not sure from the text/methods how they are defining source or sink sites. I think this is more about dispersal potential and I'd recommend reworking the text around clarified definitions.

There is a difference between metapopulation source-sink dynamic concepts and our definition. Our definition of source-sink dynamics was based only on the “net flux of mosquitoes”, computed as the difference between the outflow and inflow of mosquitoes. A source by our definition is a municipality that sends more mosquitoes than it receives. A sink is a municipality that receives more mosquitoes than it sends.

However, as noted above, we have decided to move this entire source-sink analysis to another paper because all of it relies on our estimation of a link between mosquito prevalence (i.e. *Mosquito Alert* probabilities) and car transport probabilities – and we make that link clear in a paper (currently under review with *Scientific Reports*) reporting a study in which we actually sampled cars on the road and collected mosquitoes in a number of them.

The idea is interesting - but for example, I wasn't clear on how to interpret contextualize/assess uncertainty) the conclusion that Madrid is a 'net sink' for mosquitoes when presence there hasn't been confirmed.

We appreciate this comment and have clarified the issue, but moved the text to a different paper.

Minor:

4. Line 70. I'd remove the term 'back-of-the-envelope' and instead refer to it as an estimate.

We agree with the referee and have changed the sentence accordingly (see lines 111-19).

5. Line 123. The use of 'fitness' in this sentence is awkward. Perhaps the 'value' of citizen science data?

We agree with the referee and have reformulated the sentence and added a new reference related to the concept of “fitness for use” in lines 300-305.

5. Line 222. Perhaps replace 'world' with global.

Done – now on lines 406-08.

6. Line 242. Quantify 'Most'.

This was also raised by reviewer #1 (point 5) and we have added more on this issue at lines 129-49 (see also text quoted above in answer to reviewer #1, question 4). We confirm that *Mosquito Alert* reports triggered ovitrap deployment in at least 11 instances, covering the regions of Andalusia (5)¹, Murcia (4), Aragon (1) 27, and Catalonia (1). We suspect additional instances in Valencia and the Balearic Islands. Quantifying this is difficult, however, because tiger mosquito surveillance and control is done at many administrative levels and from different sectors (municipalities, provinces, regions, central government, universities, private companies, etc.) without a common working framework.

7. Line 276. Why were half-Cauchy priors chosen (e.g., versus uniform or inverse gamma)?

The half-Cauchy is recommended as a weakly informative prior distribution for logistic regression models by Gelman et al. (2008)⁷. The authors of that article suggest that uniform priors are often too conservative and that the half-Cauchy, while also very conservative when used with an appropriate scaling parameter, outperforms Gaussian and Laplace alternatives. In addition, in previous work, Gelman (2006)⁸ notes serious problems with the inverse-gamma as a prior.

8. Lines 313-315. I don't understand the purpose or method of resampling data or the weights inverse to participation time. Is this just a method to normalize the sampling effort variable?

We have tried to better explain this in the new draft, now at lines 532-40. The purpose is to reduce bias and variance given the high degree of imbalance in the data. The potential for problems in the absence of resampling is stressed by, for instance, Ho et al. (2007)⁹, and it became clear when trying to fit the model without resampling (e.g. lack of mixing among MCMC chains). Our resampling approach is similar to that proposed by Ho and colleagues (2007)⁹ and can be seen as a type of non-parametric pre-processing in which matching/resampling is done based on the independent variable. We have also included a new Supplementary Figure 2, showing the data before and after resampling.

References:

1. Delacour-Estrella, S. *et al.* Primera cita de mosquito tigre, *Aedes albopictus* (Diptera, Culicidae), para Andalucía y primera corroboración de los datos de la aplicación Tigatrapp/First record of. *An. Biol.* 93–96 (2014). doi:10.6018/analesbio.36.16
2. Delacour-Estrella, S. *et al.* Primera cita del mosquito invasor *Aedes albopictus* (Diptera, Culicidae) en Aragón: confirmación de su presencia en huesca capital. *Boletín la Soc. Entomológica Aragon.* **58**, 157–158 (2016).
3. Alarcón-Elbal, P. M. *et al.* Updated distribution of *Aedes albopictus* (Diptera: Culicidae) in Spain: new findings in the mainland Spanish Levante, 2013. *Mem. Inst. Oswaldo Cruz* **109**, 782–786 (2014).
4. Vehtari, A., Gelman, A. & Gabry, J. Practical Bayesian model evaluation using leave-one-out cross-validation and WAIC. *Stat. Comput.* 1–20 (2016). doi:10.1007/s11222-016-9696-4
5. Oltra, A., Palmer, J. R. B. & Bartumeus, F. AtrapaelTigre.com: Enlisting Citizen-Scientists in the War on Tiger Mosquitoes. In *European Handbook of Crowdsourced Geographic Information* (eds. Capineri, C. et al.) 295–308 (Ubiquity Press, 2016). doi:http://dx.doi.org/10.5334/bax
6. Lazer, D., Kennedy, R., King, G. & Vespignani, A. The Parable of Google Flu: Traps in Big Data Analysis. *Science (80-)*. **343**, 1203–1205 (2014).
7. Gelman, A., Jakulin, A., Pittau, M. & Su, Y. A weakly informative default prior distribution for logistic and other regression models. *Ann. Appl. Stat.* (2008).
8. Gelman, A. Prior distributions for variance parameters in hierarchical models. 515–533 (2006).
9. Ho, D. E., Imai, K., King, G. & Stuart, E. A. Matching as nonparametric preprocessing for reducing model dependence in parametric causal inference. *Polit. Anal.* **15**, 199–236 (2007).

Reviewers' comments:

Reviewer #1 (Remarks to the Author):

The major revision of the manuscript has improved the quality of this work quite a lot.

A couple of minor points:

- I would move the sentence in parenthesis in line 63 to the Discussion Section commenting on future work.
- Do you have any reference supporting the claim in line 83 about detection coming from the public. If not you could remove this as it seems unsupported before your article.
- In the legend of Fig 3. it is not clear what the difference is between Mosquito Alert alone and in combination with ovitraps and ovitraps alone and in combination with Mosquito Alert. Is it the order of detection?
- The same is true for the paragraph in line 150. Mosquito Alert alone had 108 detections and is credited with 175 detections. Is the 67 difference because Mosquito Alert detected it first and then it was confirmed by ovitrap? It would be good to clarify this.
- Line 157: what does "the invasion front is contiguous" mean. Please define it.
- It would be helpful to define area under the curve (AUC) in the legend of Fig. 6.
- Regarding the references: Remove (London, England) from the journal name in 7. Provide the URL in 8. Provide number and pages 12, unless it is a book then provide publisher else provide an URL. Same for 16 and 53. Remove (80-) from the journal name in 37. Provide URLs for the software references.
- Define N in Supplementary Table 1

Reviewer #2 (Remarks to the Author):

The revised manuscript has greatly improved coherence and logical flow and was a pleasure to read. I remain interested in how the sensitivity of mosquito alert differs between established and 'edge' municipalities. It may be great at picking up albopictus (with high sensitivity) when sensitivity is estimated across all regions but this doesn't answer the primary question about early invasion detection. The methods and equation (1, line 409) describe a sensitivity analysis for the total area. It isn't immediately clear from the results or methods whether or not sensitivity was calculated just for the invasion front (274 municipalities) or the broader area considered. For example, were the 112 municipalities (line 186) all first detections in 2015?

I think that there is a lot of interesting information in the results lines 150-175 that is related to this question. The associated figures are great but the text may be a bit longer and more difficult to follow than needed. This text shows that Mosquito Alert recorded first detection across the same total area (of the municipalities with new reports) as ovitraps. But the ideas in the next paragraph (lines 157-163) aren't clear. Is it saying that Mosquito Alert participants are only reporting from 25% of the 274 first-time municipalities (vs 50% covered by ovitraps)? Likewise, the message in the next paragraph about median and mean distances to the front could be clarified. Presumably both mean and medians are presented to make a point – but it isn't clear (to this reader) what it is.

Finally, while it isn't entirely necessary for the goals of this paper – i.e., introducing a methods approach- it would be useful to mention the challenges with detection and citizen engagement (e.g., cell phone/app use) across the varied socio-ecological neighborhoods that are known to significantly affect *Aedes* species abundances and exposures (for example: (Honorio et al. 2009, Unlu et al. 2011, LaDeau et al. 2013, Ali et al. 2017). Ovitrap are also likely to be biased towards deployment in neighborhoods where people have resources – so this point isn't a challenge to the direct comparisons being made.

Ali, S., O. Gugliemini, S. Harber, A. Harrison, L. Houle, J. Ivory, S. Kersten, R. Khan, J. Kim, C. LeBoa, E. Nez-Whitfield, J. O'Marr, E. Rothenberg, R. M. Segnitz, S. Sila, A. Verwillow, M. Vogt, A. Yang, and E. A. Mordecai. 2017. Environmental and Social Change Drive the Explosive Emergence of Zika Virus in the Americas. *Plos Neglected Tropical Diseases* 11.

Honorio, N. A., R. M. R. Nogueira, C. T. Codeco, M. S. Carvalho, O. G. Cruz, M. Magalhaes, J. M. G. de Araujo, E. S. M. de Araujo, M. Q. Gomes, L. S. Pinheiro, C. D. Pinel, and R. Lourenco-de-Oliveira. 2009. Spatial Evaluation and Modeling of Dengue Seroprevalence and Vector Density in Rio de Janeiro, Brazil. *Plos Neglected Tropical Diseases* 3.

LaDeau, S. L., P. T. Leisnham, D. Biehler, and D. Bodner. 2013. Higher mosquito production in low-income neighborhoods of Baltimore and Washington, DC: understanding ecological drivers and mosquito-borne disease risk in temperate cities. *Int J Environ Res Public Health* 10:1505-1526.

Unlu, I., A. Farajollahi, S. P. Healy, T. Crepeau, K. Bartlett-Healy, E. Williges, D. Strickman, G. G. Clark, R. Gaugler, and D. M. Fonseca. 2011. Area-wide management of *Aedes albopictus*: choice of study sites based on geospatial characteristics, socioeconomic factors and mosquito populations. *Pest Management Science* 67:965-974.

Detailed Responses to Reviewers

General Points:

We appreciate very much the reviewers' latest suggestions. We agree with all of them and have addressed them as follows:

Reviewer #1:

1. I would move the sentence in parenthesis in line 63 to the Discussion Section commenting on future work.

We have done this, moving it to lines 351-52 of the Discussion Section. We have also added related information on our recent work with the U.N. to create the *Global Mosquito Alert* initiative, targeting disease-vector mosquitoes worldwide.

2. Do you have any reference supporting the claim in line 83 about detection coming from the public. If not you could remove this as it seems unsupported before your article.

We have added the following references (line 84 of the revised version), which show how public complaints led to the first detection of tiger mosquitoes in Spain (Aranda *et al.* 2006), Albania (Adhami & Reiter 1998), and Algeria (Benallal *et al.* 2016), and how public complaints often lead to new detections within Spain (Alarcón-Elbal *et al.* 2014):

Aranda, C., Eritja, R. & Roiz, D. First record and establishment of the mosquito *Aedes albopictus* in Spain. *Med. Vet. Entomol.* **20**, 150–2 (2006).

Adhami, J. & Reiter, P. Introduction and establishment of *Aedes (Stegomyia) albopictus* skuse (Diptera: Culicidae) in Albania. *J. Am. Mosq. Control Assoc.* **14**, 340–3 (1998).

Alarcón-Elbal, P. M. *et al.* Updated distribution of *Aedes albopictus* (Diptera: Culicidae) in Spain: new findings in the mainland Spanish Levante, 2013. *Mem. Inst. Oswaldo Cruz* **109**, 782–786 (2014).

Benallal, K. E., Allal-Ikhlef, A., Benhamouda, K., Schaffner, F. & Harrat, Z. First report of *Aedes (Stegomyia) albopictus* (Diptera: Culicidae) in Oran, West of Algeria. *Acta Tropica* **164**, (2016).

2. In the legend of Fig 3. it is not clear what the difference is between Mosquito Alert alone and in combination with ovitraps and ovitraps alone and in combination with Mosquito Alert. Is it the order of detection?

We are not referring to the order of detection – we simply phased this in a way that was not clear. What we mean by “Mosquito Alert alone and in combination with ovitraps” is the set of all municipalities in which Mosquito Alert is alone credited with the detection and all those in which both sources are credited. Similarly, “ovitraps alone and in combination with Mosquito Alert” is the set of all municipalities in which ovitraps are alone credited and all those in which both sources are credited. For both sets, “all those in which both sources are credited” are the same municipalities. The reason for these categories is that they capture the full distributions of detections by each source. We could simply say “all municipalities in which Mosquito Alert is credited” and “all municipalities in which ovitraps are credited” but we worry that doing so would not make clear that these two sets overlap. We have tried to clarify this in the legend.

3. The same is true for the paragraph in line 150. Mosquito Alert alone had 108 detections and is credited with 175 detections. Is the 67 difference because Mosquito Alert detected it first and then it was confirmed by ovitrap? It would be good to clarify this.

We have tried to clarify this as well (lines 150-57 of the revised version). Again, it is not the order but simply whether the overlapping municipalities are included.

4. Line 157: what does "the invasion front is contiguous" mean. Please define it.

We are referring here to municipalities that lie along the known invasion front – i.e., share a border with a municipality in which tiger mosquitoes had been previously detected. We have tried to make this paragraph clearer in the revised version (lines 159-69).

5. It would be helpful to define area under the curve (AUC) in the legend of Fig. 6.

Done.

6. Regarding the references: Remove (London, England) from the journal name in 7. Provide the URL in 8. Provide number and pages 12, unless it is a book then provide publisher else provide an URL. Same for 16 and 53. Remove (80-) from the journal name in 37. Provide URLs for the software references.

Done.

7. Define N in Supplementary Table 1

Done.

Reviewer #2:

1. The revised manuscript has greatly improved coherence and logical flow and was a pleasure to read. I remain interested in how the sensitivity of mosquito alert differs between established and ‘edge’ municipalities. It may be great at picking up albopictus (with high sensitivity) when sensitivity is estimated across all regions but this doesn’t answer the primary question about early invasion detection. The methods and equation (1, line 409) describe a sensitivity analysis for the total area. It isn’t immediate clear from the results or methods whether or not sensitivity was calculated just for the invasion front (274 municipalities) or the broader area considered. For example, were the 112 municipalities (line 186) all first detections in 2015?

This is a fair point. We had included the known invasion area in these calculations, so the 112 municipalities were not all first detections. Our thinking had been that the country-wide calculation should simply include all municipalities, but your approach is clearly more consistent with the idea of early warning and the inclusion of the known invasion area may bias the result in favor of *Mosquito Alert* given that ovitraps tend not to be placed in that area. We have now re-done these calculations using three approaches: (1) all of Spain including known invasion area (as before), (2) all of Spain excluding known invasion area, and (3) only municipalities within 30 km of known invasion area (and excluding that area itself). In addition, for all of these calculations we now use data from both 2014 and 2015 (whereas we had previously restricted it to 2015). As explained in the revised version, method (1) gives a ratio of 1.01, method (2) gives 2.07 and method (3) gives 1.35. (In all cases, the ratios drop if we increase the estimated specificity of *Mosquito Alert*.) The choice of method really hinges on how one views the sensitivity question and whether one assumes that the inclusion of the known invasion area biases the results in favor of *Mosquito Alert*. By presenting all of them we can at least place some boundaries on the true ratio.

2. I think that there is a lot of interesting information in the results lines 150-175 that is related to this question. The associated figures are great but the text may be a bit longer and more difficult to follow than needed. This text shows that Mosquito Alert recorded first detection across the same total area (of the municipalities with new reports) as ovitraps. But the ideas in the next paragraph (lines 157-163) aren’t clear. Is it saying that Mosquito Alert participants are only reporting from 25% of the 274 first-time municipalities (vs 50% covered by ovitraps)? Likewise, the message in the next paragraph about median and mean distances to the front could be clarified. Presumably both mean and medians are presented to make a point – but it isn’t clear (to this reader) what it is.

The point of both of these paragraphs is to show how *Mosquito Alert* detections tend to be farther from the known invasion area than ovitrap detections. We have tried to clarify this in the revised version.

3. Finally, while it isn’t entirely necessary for the goals of this paper – i.e., introducing a methods approach- it would be useful to mention the challenges with detection and

citizen engagement (e.g., cell phone/app use) across the varied socio-ecological neighborhoods that are known to significantly affect Aedes species abundances and exposures (for example: (Honorio et al. 2009, Unlu et al. 2011, LaDeau et al. 2013, Ali et al. 2017). Ovitrap are also likely to be biased towards deployment in neighborhoods where people have resources – so this point isn't a challenge to the direct comparisons being made.

This is a really interesting point and we have now included discussion of these papers in the Discussion section as well as citing them in the beginning of the main text (lines 78-80, 292-99, 319-27).

REVIEWERS' COMMENTS:

Reviewer #1 (Remarks to the Author):

All comments in my review have been satisfactorily addressed.

Reviewer #2 (Remarks to the Author):

The authors have done a good job addressing the previous comments and this paper is greatly improved in clarity.